# Cryptanalysis and improved mutual authentication key agreement protocol using pseudo-identity

**Hyang-Rim Jo**◉**, Kyong-Sok Pak**◉◉***, Chung-Hyok Kim**◉**, Il-Jin Zhang**◉

Faculty of Information Science, Kim Il Sung University, Pyongyang, the Democratic People's Republic of Korea

◉ These authors contributed equally to this work.

* ks.pak0228@ryongnamsan.edu.kp

**Data Availability Statement:** All relevant data are within the paper.

**Funding:** The authors received no specific funding for this work.

## Abstract

The authentication key agreement is a scheme that generates a session key for encrypted communication between two participants. In the authentication key agreement, to provide the mutual authentication and the robust session key agreement is one of the important security requirements to enhance the security performance of key agreement. Recently Zhou et al. had proposed the key agreement protocol using pseudo-identifiers, but we found that there were weaknesses in their protocol. We have demonstrated that Zhou et al.'s protocol is vulnerable to replay attack, fails to provide mutual authentication, no key control, re-registration with the original identifier and efficiency in the verification of wrong password. We improved their scheme and proposed an improved authentication key agreement protocol that provides robust mutual authentication and the secure session key agreement. We analyzed its security performance using BAN logic and AVISPA tools and compared computational cost, communication overhead and security properties with other related schemes.

## 1. Introduction

Authentication key agreement(AKA) is one of the important issues to ensure the confidentiality of network security (to protect user privacy and network resources) as a scheme where the session key is exchanged to encrypt a message exchanged between communication participants on a public network. The authentication key agreement protocol can be divided into key agreement for end-to-end communication, and key agreement for end-to-server communication, depending on the entities involved in the communication. In the key agreement protocol for end-to-end communication, two parties participating in key exchange are both users and it applies for the encrypted communication between users. The key agreement protocol for end-to-server communication is used for encrypted communication between the user and several servers or service providers. Key agreement for end-to-server communication can be classified as a key agreement scheme (SS-AKA) between single server and end users, a key agreement scheme between multiple servers and end users (MS-AKA) [1–38]. Recently, with the introduction of technologies such as P2p, cloud computing, WSN, and IoT, researchers are further

**Competing interests:** The authors have declared that no competing interests exist.

investigating authentication key agreement between end-to-end servers. Particularly, research on key agreement for communication between multiple servers and end users(MS-AKA) is focused on.

For the MS-AKA implementation, researchers introduced multi-factor authentication such as password, smart card and biometrics and they used public key cryptographics and non-cryptographics for key agreement. Research has mainly focused on lightweight and security enhancement. The research for lightweight is based on non-cryptographics [1–13, 24–32] that only uses hash function and XOR operation, and the research for security performance improvement is mainly based on public key encryption [14–21, 32–37].

However, most of the lightweight approaches suffer from low security performance, and security performance improvements suffer from high computational costs and communication overhead. Research on authenticated key agreement scheme with higher security and lower computational cost and lower communication overhead is still a challenge for researchers. In particular, user anonymity and mutual authentication are very important properties for authenticated key agreement. In this paper, we propose an improved authenticated key agreement scheme based on pseudo-identiy and chaotic maps to provide user anonymity and mutual authentication.

## 1.1 Related work

In order to implement authentication key agreement between multiple servers and end users, researchers have studied both the key agreement scheme [1–21] where the registration center does not participate in key exchange and the key agreement scheme [24–38] where the registration center participates in key exchange.

In such a way that the registration center does not participate in key exchange, users, servers, or service providers register on the registration center in the system registration phase, and in the key exchange phase, exchange the key without the involvement of the registration center.

Research has mainly been done in terms of security performance enhancement rather than lightening of computational cost.

As authentication factors for the user, they used passwords, smart cards, and biometric information, and used a pre-shared key, a group key and secret-sharing technique for authentication to the service system.

The researchers used computationally efficient hash functions, elliptic curve cryptosystem (ECC), and Chebyshev chaotic maps (CCM) for key agreement to enhance the security performance of key exchange schemes between multiple servers and end users. For key agreement, they used hash functions and performed user authentication using a dynamic identifier and a pre-shared key. In 2007, Liao et al. [1] proposed a secure dynamic identifier-based remote user authentication scheme in a multi-server environment. But it was revealed that his protocol is vulnerable to insider attacks, impersonation attacks, server spoofing attacks, registration center spoofing attacks, and fails to provide mutual authentication by Hsiang et al. [22] in 2009. In 2012, Li et al. [2] proposed a new remote user authentication scheme based on smart card and dynamic identifier for multi-server environments. In order to protect the user identifier from tracking, their scheme allows the user's identifier to change dynamically whenever the user logs on to the server. In 2012, Tsaur et al. [3] proposed an efficient and secure multi-server authentication scheme with key exchange. However, his protocol was found to be vulnerable to offline password guessing attacks, privileged insider attacks, and malicious user attacks by Xu et al. [4] in 2013. Xu et al. proposed a new dynamic identification-based authentication scheme for a multi-server environment using smart cards. He proposed an improved dynamic

identity based scheme to eliminate all the security and efficiency weaknesses without decreasing other security performances in his work. In 2014, Chuang et al. [5] proposed an anonymous multi-server authenticated key agreement scheme based on trust computing using smart cards and biometrics. However, it was revealed that his protocol is vulnerable to denial of service attacks, stolen smartcard attacks, user impersonation attacks, and server spoofing attacks by Maitra et al. [6] and Mishra et al. [7] in 2014. Maitra et al. proposed efficient remote user authentication using biometric and password-based smart cards for telemedicine information systems in a multi-server environment. He found that an unregistered attacker can successfully log into the system as a valid user in Chuang et al.'s scheme, and in order to overcome vulnerabilities, he proposed a scheme that allows users to register simultaneously on a root remote server called registration center to be served from all branch remote servers using a registered smart card. In 2014 Mishra et al. proposed a multi-server authentication key agreement scheme using smart cards based on biometrics to preserve secure user anonymity. He improved Chuang et al.'s protocol, but it was found that his protocol is vulnerable to impersonation attacks, replay attacks, denial of service attacks, fails to achieve perfect forward security and no user re-registration phase by Wang et al. [10] in 2016. Wang et al. proposed cryptanalysis and improvement on multi-server authentication and key agreement schemes based on biometrics. But it was revealed that his protocol is vulnerable to user impersonation attacks, privileged insider attacks, and server impersonation attacks and does not provide perfect forward security by Yang et al. [20] in 2018. Yang et al. designed a protocol that performs mutual authentication between the user and the service provider and exchanges key without involvement of the registration center in a multi-service system environment. In his protocol, the registration center shares the pre-shared key (PSK) and long term key with service providers. In 2015, Amin et al. [8] proposed a new user authentication and key agreement protocol for multiple healthcare provider access available in TMIS. They developed a new structure for access to multiple healthcare providers in order to decrease the vulnerability of a single healthcare provider, where the user can communicate directly and safely with the doctor of the healthcare provider. They also developed smart card-based authentication and key agreement security protocols that can be used in TMIS systems using one-way hash functions as cryptography. In 2017, Guo et al. [11] proposed a key exchange protocol that provides user anonymity in a multi-service system environment. In key exchange, the registration server does not participate, shares a pre-shared key with the service providers and uses the public key of the service providers.

In 2019 Lwamo et al. [12] proposed a key exchange scheme without a third-party server using hash functions and symmetric key encryption. He demonstrated lightweight and anonymity, and the user identifier is encrypted with the service provider's public key, and it is updated every round. In key exchange using only hash functions, the registration center shares pre-shared key with the service providers to authenticate them. In 2020, Mishra et al. [13] proposed a dynamic ID-based authenticated key agreement scheme for mobile edge computing without a trusted third party. The proposed scheme guarantees mutual authentication between user and edge servers and achieves important security properties such as secure communication, mutual authentication, user anonymity, and session key agreement.

To overcome the disadvantage of using a pre-shared key, researchers used public-key encryption for key exchange.

In 2014, Han et al. [14] proposed an identifier-based mutual authentication with a key agreement protocol for a multi-server environment based on elliptic curve cryptography. In order to improve the performance of precedent bilinear pairing-based several authentication schemes in a multi-server environment, they proposed a new identifier-based mutual authentication protocol using signature based elliptic curve cryptography. In 2016, Chaudhry et al. [9]

proposed a secure biometrics-based multi-server authentication scheme for social multimedia networks. They show that first one of the two schemes of Lu et al. [37] designed for multi-server architectures is vulnerable to impersonation attacks and doesn't provide user anonymity, and the second one is vulnerable to user impersonation attacks. They proposed an enhanced scheme, and used elliptic curve cryptography and hash functions for key exchange. In 2019, Ying et al. [15] proposed a lightweight remote user authentication protocol for multi-server 5G networks using self-verified public-key encryption. To reduce the computational complexity, they used self-verified public-key cryptography based on elliptic curve cryptography to verify the valid of users and servers. Without pairing operations, their scheme could improve computational efficiency and provide mutual authentication. In 2016, Irshad et al. [16] proposed an anonymous multi-user authentication key exchange protocol based on chaotic mapping using smart cards. They reviewed recent multi-server authentication schemes and proposed a single-round trip multi-server authentication protocol based on chaotic mapping to overcome their schemes' limitations. In 2017 Kumari et al. [17] proposed a user key exchange scheme in a multi-server environment using chaotic mapping. His scheme is based on a single-sign-on, where the registration center pre-shares the secret key with the service providers. In 2019 Qiao et al. [18] proposed an authentication key exchange scheme that provides strong anonymity for multi-server environments in TMIS. His scheme exchanges key with chaotic mapping and encrypts the identifier with symmetric encryption without involving the key exchange server. In 2012 Chuang et al. [19] proposed a generalized identifier-based user authentication scheme for a mobile multi-server environment. In his work, he first proposed a security model for a multi-server environment and then a bilinear pairing-based mutual authentication and key exchange scheme. Their scheme can be used for both common users with long valid periods and anonymous users with short valid periods. In 2020 Yu et al. [21] proposed a key agreement scheme (AKA-NS) that shares keys without authentication servers in an IoT-based cloud environment based on bilinear pairing. His scheme authenticates users based on the Elgamal cryptography signature, and uses secret values based on bilinear pairingand hash functions for key agreement.

To overcome the disadvantage of using a pre-shared key, researchers also proposed a combination of group key agreement and secret-sharing techniques [22, 23]. In 2021, Vinoth et al. [22] proposed a secure multifactor authenticated key agreement scheme for industrial IoT environment to support authorized user remotely accessing the sensors. In their proposed scheme, only hash functions, XOR operation and symmetric encryption are used for session key agreement, and sensor devices share secret information by combining group key agreement and secret-sharing techniques.

The key agreement scheme involving the registration center in key exchange has been studied towards lightweight rather than maintaining security performance. Researchers used passwords, smart cards, and biometric information as the authentication factors for the user, and used a pre-shared key for authentication of the service provider system. In order to lighten the computational cost of the scheme, researchers focused on computationally efficient hash functions, ECC and ECM for key agreement. Some researchers designed the protocol where the registration center participates in key exchange, using only a hash function without using public-key encryption in key exchange to reduce computational cost.

In 2009, Hsiang et al. [24] proposed a secure dynamic ID-based remote user authentication scheme for a multi-server environment. But it was revealed that his protocol is vulnerable to impersonation attacks, server spoofing attacks, cannot be easily repaired, and cannot provide mutual authentication by Lee et al. [25] In 2011, Lee et al. proposed a secure dynamic identifier-based remote user authentication scheme for a multi-server environment using smart cards. But it was revealed that his protocol is vulnerable to impersonation attacks and server

spoofing attacks, and if the mutual authentication message is partly modified by the attacker, it cannot provide a corresponding authentication by Li et al. [2] in 2012. In 2014, Xue et al. [26] proposed a lightweight dynamic anonymous identity-based authentication and key exchange protocol that does not use verification tables in a multi-server environment. However, in 2015, Gupta et al. [27] has shown that Xue et al.'s protocol is vulnerable to known password guessing attacks, stolen smartcard attacks, and impersonation attacks, and in 2018, Amin et al. [29] found that Xue et al.'s protocol has flaws in user anonymity, offline password guessing attacks, privileged insider attacks, no key control, user impersonation attacks. Gupta et al. proposed a hash function-based multi-server key exchange protocol with smart cards. But in 2019, it was found that his protocol is vulnerable to denial of service attack, stolen smart card attack, and user impersonation attacks and that it does not achieve perfect forward security by Tomar et al. [35]. Tomar et al. proposed an authentication key exchange protocol with a password, biometrics (Fuzzy extractor) and smart cards. His protocol uses timestamps, uses elliptic curve cryptography to establish the session key, and performs mutual authentication using two control servers. Amin et al. proposed the anonymous authentication and key exchange protocol between a user and multi-server in cloud environment. In his work, the server and user use a shared secret that combines the server's secret with the user's identifier, and the user accesses to the smart card using the password. In 2016, Maitra et al. [28] proposed an enhanced multi-server authentication protocol using passwords and smart cards. He found that some flaws in the precedent works, and he proposed a new protocol, focusing on the improvement of their security performances, and used symmetric key encryption. In 2018, Wei et al. [30] proposed a two-factor authentication key exchange protocol using the password and secret keys stored in smart cards in cloud environments. They used the shared-secret key combined with time-stamps as a message-encryption key. In 2019, Zhou et al. [31] proposed a lightweight authentication key exchange protocol based on a hash function in cloud computing environment. In his work, they updated pseudo-identities of two participants every round. The user registers with identifiers, pseudo-identifiers, passwords, and random numbers, and the IoT controller registers with identifiers, pseudo-identifiers, and random numbers.

To enhance the security performance of key exchange, some researchers have used public key encryption such as ECC and ECM.

In 2010, Yoon et al. [32] proposed a robust multi-server authentication scheme based on biometrics using smart cards in elliptic curve cryptography. They proposed an authentication scheme without a verification table, and the proposed scheme can provide stronger user authentication by using biometrics, and provide more secure key exchange scheme based on ECC. In 2017, Chandrakar et al. [33] proposed a key exchange protocol for remote user authentication that provides three factors authentication and anonymity using elliptic curve cryptography in a multi-server environment. For the exchanged key, they use Elliptic Curve Diffie–Hellman (ECDH), but for the encryption, they use the addition of a point on elliptic curve, and XOR without the use of special encryption. In 2018, Qi et al. [34] proposed a key exchange scheme using elliptic curve cryptography in a multi-server environment. They used the server's public-key-based symmetric key encryption for the communication between the user and the server. They also used the registration center's public-key-based symmetric key encryption for communication between the server and the registration center. Thus, his protocol provides a relatively strong key exchange scheme. In 2017, Irshad et al. [36] proposed a new user authentication key exchange protocol based on chaotic mapping for a multi-server environment. They used password, biometrics, smart card and the secret key shared with the registration center to authenticate the user and used chaotic mappings and bio-hash functions to exchange the session key. In 2021, Xia [38] proposed a modular exponention based anonymous authentication and key agreement scheme with privacy-preserving in IoT environment

for smart city, and the work for authenticated key agreement scheme was studied not only in P2P, IoT environment, but also in VANET environment [39].

## 1.2 Motivation and our contribution

According to the research of the precedent schemes, we found that key agreement protocols without the registration server have several disadvantages such as the mutual authentication, anonymity and untraceability in their implementation for communication between multiple servers and end users [1, 3, 7, 10]. Also we found that the research in protocols with the registration server has been intensified towards lightweight, but on the other hand, their security performance has become weakened [24, 26, 29]. In our work, we analysed the pre-shared key-based Lwamo et al.'s scheme [12] where the registration center doesn't participate in the key agreement, and found that his scheme is vulnerable to the stolen smart card attack. We also analysed the pseudo-identities-based Zhou et al.'s scheme [31] where the registration center participates in the key agreement, and found that their scheme is vulnerable to replay attack and does not provide mutual authentication, no key control, re-registration with an original identity, and efficiency in the verification of wrong password. From this research, we propose an improved authentication key agreement protocol for communication between multi-servers and end users to overcome the flaws of Zhou et al.'s scheme. Finally, we analysed the security properties of our protocol and performed comparative analysis with precedent protocols to show that our protocol is superior in terms of security properties and computational complexity.

## 2. Preliminaries

This section describes Fuzzy extractor, Chebyshev chaotic maps, their computational problems and threat model.

### 2.1 Fuzzy extractor

The fuzzy extractor includes two functions *Gen* and *Rep*. The function *Gen* extracts biometric input *BI*, and outputs a nearly random binary string *R* and an auxiliary binary string *P*. And the function *Rep* recovers *R* with the assistance of corresponding auxiliary string *P* and biometric $BI^*$. If dis($BI, BI^*$) ≤ t and $Gen(BI) \rightarrow <R, P>$, then we have $Rep(BI^*, P) = R$. Otherwise, there is no guarantee provided by function *Rep*. The literature [40, 41] describes more details about the fuzzy extractor.

### 2.2 Chebyshev polynomials

Chebyshev polynomial $T_m(a)$ is defined as follows [42].

$$T_m(a) = cos(m \cdot arcos(a)), \ a \in [-1, 1], m \in N$$

Chebyshev polynomials satisfy the following recursive relationship [42].

$$T_m(a) = 2a \cdot T_{m-1}(a) - T_{m-2}(a)(m > 2),$$

$$T_0(a) = 1, \ T_1(a) = a$$

### 2.3 The property of Chebyshev polynomials

Chebyshev polynomials have the following two properties [42, 43].

Chaotic property: When $m>1$, Chebyshev polynomial map $T_m(a)$: $[-1,1] \rightarrow [-1,1]$ of degree $m$ is a chaotic map with its invariant density $f^*(a) = \frac{1}{\pi\sqrt{1-a^2}}$, for positive Lyapunov exponent $ln(m) > 0$.

Semi-group property: For $x, y \in N$ and any $a \in [-1,1]$, $T_x(T_y(a)) = T_{xy}(a) = T_y(T_x(a))$.

## 2.4 Enhanced Chebyshev polynomials

The semi-group property holds for Chebyshev polynomials on the interval $(-\infty, +\infty)$, which can enhance the property as follows [43]:

$$T_m(a) = 2a \cdot T_{m-1}(a) - T_{m-2}(a) \bmod p (m \geq 2, \ a \in (-\infty, +\infty), p \text{ is a large prime number}),$$

$$T_x(T_y(a)) \equiv T_{xy}(a) \equiv T_y(T_x(a)) \bmod p (x, y \in N).$$

## 2.5 Computational problems based on Chebyshev polynomials

CDLP (Chaotic map-based Discrete Logarithm problem): For given two real numbers $a$ and $b$, it is infeasible to find the integer $m$ by any polynomial time bounded algorithm, where $b = T_m(a) \bmod p$ [43].

CDHP (Chaotic map-based Diffie-Hellman problem): For given three elements $a$, $T_x(a) \bmod p$ and $T_y(a) \bmod p$, it is infeasible to compute the value $T_{xy}(a) \bmod p$ by any polynomial time bounded algorithm [43].

## 2.6 Threat model

In this subsection, we introduce several threat models including the Dolev-Yao threat model [44], side channel attack [45], and password guessing attack [46], for the security analysis of the proposed scheme and previous schemes.

1. An attacker can eavesdrop, modify, remove, block and retransmit all messages transmitted on the public channel [44].

2. An attacker can extract all stored data from a lost or stolen smart card as a power analysis attack [40].

3. An attacker can perform offline and online password guessing attacks after obtaining information from user's smart card [46].

4. An attacker can be a malicious user or an outside hacker [20].

## 3. Analysis of precedent schemes

In this section, we review the schemes proposed by Lwamo et al. [12] and Zhou et al. [31], and show that their schemes have some flaws.

### 3.1 Analysis of Lwamo et al.'s scheme

**3.1.1 Lwamo et al.'s scheme.** Lwamo et al. proposed the authentication key agreement protocol without the registration center using hash function and symmetric key encryption. The user identity is encrypted with the public key of the service server and it is updated every round.

Table 1 shows the notations used in his scheme.

**Table 1. Notations in Lwamo et al.'s scheme.**

| Notation | Description |
|---|---|
| $A_i$ | The $i$th user |
| $B_j$ | The $j$th server |
| $S$ | The secret value of $RC$ |
| $RC$ | The registration center |
| $U_i$ | $A_i$'s identity |
| $MU_i$ | $A_i$'s masked identity |
| $S_j$ | $B_j$'s identity |
| $MS_j$ | $B_j$'s masked identity |
| $P_i$ | $A_i$'s password |
| $MP_i$ | $A_i$'s masked password |
| $BO_i$ | $A_i$'s biometric information |
| $h(.)$ | One-way hash function |
| $VPSK$ | A secure pre-shared key between $RC$ and the server |
| $a_i$ | A nonce |
| $pu_j$ | $B_j$'s public key |
| $pr_j$ | $B_j$'s private key |
| $Ek()$ | Encryption with $k$ as a key |
| $Dk()$ | Decryption with $k$ as a key |
| $\oplus$ | XOR operator |
| $\|$ | concatenation operator |

### Registration phase

- Registration for server
  To be a valid server, the server sends a registration request to $RC$ via a secure channel. The server's identity $S_j$ and its public key $pu_j$ are contained in the registration request. Then $RC$ sends $VPSK$ and $s$ to the server by a response via a secure channel like Internet Key Exchange version 2 (IKEv2) and publishes $pu_j$.

- Registration for user

**Step 1**: First the user $A_i$ selects his/her identity $U_i$, password $P_i$, a nonce $a_i$ and biometric information $BO_i$.

**Step 2**: $A_i$ computes $MP_i = h(U_i \| a_i \| P_i)$ and $VREG = h(MP_i \oplus BO_i)$.

**Step 3**: The registration request $\{U_i, VREG = h(MP_i \oplus BO_i)\}$ is sent to $RC$ by $A_i$.

**Step 4**: $SA_i = h(U_i \| s)$, $SB_i = h(SA_i)$ and $SC_i = h(VREG = h(MP_i \oplus BO_i)) \oplus SB_i$ are generated by $RC$.

**Step 5**: The $RC$ chooses a nonce $a_{ci}$ for $A_i$, computes $MU_i = E_s(U_i \| a_{ci})$, $SD_i = VPSK \oplus MU_i$ and makes the smart card storing $\{MU_i, SB_i, SC_i, SD_i, h(.)\}$ its own possession.

**Step 6**: The $A_i$ inserts the nonce $a_i$ into the smart card, which now includes $SC = \{MU_i, a_i, SB_i, SC_i, SD_i, h(.)\}$ and owns the smart card.

### Login and authentication phase

**Step 1**: $A_i$ inserts the smart card into the reader and enters $U_i$, $P_i$ and $BO_i$.

**Step 2**: Then the smart card computes $SB_i^* = SC_i \oplus h(h(U_i||a_i||P_i) \oplus BO_i)$ and sees if $SB_i^* = SB_i$. If $SB_i^* \neq SB_i$, the smart card stops the login phase. The smart card is blocked if the two values do not match for three continued trials within limited threshold time.

**Step 3**: Then the smart card chooses a nonce $Ra_i$ and computes $M_1 = h(SB_i) \oplus Ra_i$, $M_2 = h(Ra_i || MU_i || SD_i ||T_1)$, where $T_1$ is the current time on the smart card, and $M_3 = Epu_j(MU_i, M_1)$.

**Step 4**: The smart card sends the login request $LOGIN \{M_2, M_3, S_j, T_1\}$ to the server $B_j$.

**Step 5**: When $B_j$ receives the request, the difference between the received login request time $T_1$ and the server time $T_2$ is computed as $\Delta T = T_2 - T_1$ by $B_j$, the phase is terminated by the server if the difference is bigger than the required transfer time.

**Step 6**: To get $MU_i$ and $M_1$, $M_3$ is decrypted as $Dpr_j(M_3)$ by the server. And to get the real identity of the user $MU_i$ is decrypted as $Ds(MU_i) = U_i || a_{ci}$ by the server.

**Step 7**: The server calculates $SA_i^* = h(U_i||s)$ and $R_{ai} = M_1 \oplus h^2(SA_i^*)$.

**Step 8**: $M_2^* = h(R_{ai} || MU_i || (VPSK \oplus MU_i) ||T_1)$ is computed and it is checked if $M_2 = M_2^*$ by the server. The session is stopped if $M_2 \neq M_2^*$.

**Step 9**: The server selects two nonces $Ra_j$ and $Ra_j^{new}$, calculates $s^* = h(Ra_i || T_3)$ where $T_3$ is the current server time, and computes a new identity for $A_i$ as $MU_i^{new} = h(U_i|| Ra_j^{new})$.

**Step 10**: Then a challenge message $M_4 = E_s (MU_i^{new}||Ra_j||Ra_i||U_i||R_j^{new}||S_j)$ and the masked identity $MS_j = h(S_j \oplus Ra_j)$ are computed by the server. And the message $CHALLENGE \{M_4, MS_j, T_3\}$ is sent to $A_i$.

**Step 11**: After receiving the message from the server, the smart card computes the difference $\Delta T = T_4 - T_3$, where $T_4$ is the current time on the smart card. The smart card stops the session if the difference is bigger than the defined interval.

**Step 12**: The smart card calculates $s^* = h(Ra_i || T_3)$ and $M_4$ is decrypted as $Ds^*(M_4)$ to get $MU_i^{new}$, $Ra_j$, $Ra_i$, $Ra_j^{new}$, $U_i$ and $S_j$.

**Step 13**: The smart card calculates $MU_i^{new*} = h(U_i|| Ra_j^{new*})$ and $MS_j^* = h(S_j||Ra_j)$. And then it sees if $MS_j = MS_j^*$ and if $MU_i^{new*} = MU_i^{new}$. Also the smart card sees if $R_i$ and $U_i$ are equal to those sent to the server on the login request, if they do not match, the smart card terminates the session.

**Step 14**: The response message $M_5 = h(Ra_j||MU_i^{new}||Ra_i)$ and the session key $Sk_{ij} = h(Ra_i||SB_i||S_j||Ra_j)$ is calculated by the smart card. And the smart card sends the response $RESP \{M_5\}$ to $B_j$.

**Step 15**: After the server receives the response, it calculates $M_5^* = h(Ra_j||MU_i^{new}||Ra_i)$ and sees if $M_5^* = M_5$. The session is stopped if $M_5^* \neq M_5$.

**Step 16**: The session key $Sk_{ij} = h(Ra_i||h^2(U_i||s)||S_j||Ra_j)$ is computed by the server. In this step, the mutual authentication between the user and the server is achieved and the session key between them is created.

**Password update phase**

**Step 1**: The user inserts the smart card into the card reader and inputs $U_i$, $P_i$ and $BO_i$.

**Step 2**: The smart card computes $SB_i^* = SC_i \oplus h((h(U_i||a_i||P_i) \oplus BO_i)$.

**Step 3**: The smart card sees if $SB_i^* = SB_i$ and stops the session if $SB_i^* \neq SB_i$.

**Step 4**: The smart card gets a new password $P_i^{new}$ from the user and calculates $SC_i^{new} = SC_i \oplus h$ $(h(U_i || a_i || P_i) \oplus BO_i) \oplus h(h(U_i || a_i || P_i^{new}) \oplus BO_i)$.

**Step 5**: The value of $SC_i$ is replaced with $SC_i^{new}$ by the smart card.

**3.1.2 Flaws of Lwamo et al.'s scheme.** *Stolen smart card attack.* In his scheme, $SC = \{MU_i, SB_i, SC_i, SD_i, h(.), a_i\}$ is stored in his/her smart card. Thus, an attacker can get $SB_i$ directly without inputting the identity $U_i$, the password $P_i$ or the biometric $BO_i$ if he gets the user's smart card. Then the attacker generates a random number $Ra_i$, computes $M_1 = h(SB_i) \oplus Ra_i$, $M_2 = h$ $(Ra_i || MU_i || SD_i || T_1)$ and $M_3 = E_{puj}(MU_i, M_1)$ and sends $LOGIN\{M_2, M_3, S_j, T_1\}$ with a time stamp $T_1$ to the server $B_j$. In this case, $B_j$ recognizes the attacker as a valid user $A_i$. After receiving the message $CHALLENGE \{M_4, MS_j, T_3\}$ from the server via step 10 from the step 4 of the login and authentication phase, the attacker finally gets the session key $Sk_{ij} = h(Ra_i || h^2(U_i || s) || S_j || Ra_j)$ between himself and the server $B_j$.

As a result, Lwamo et al.'s scheme is vulnerable to the stolen smart card attack.

## 3.2 Analysis of Zhou et al.'s scheme

**3.2.1 Zhou et al.'s scheme.** In 2019, Zhou et al. proposed a hash function-based lightweight authentication key agreement scheme for user-multi IoT access in IoT environment. In their scheme, the pseudo-identities are used. Table 2 shows the notations used in Zhou et al.'s scheme.

**Registration phase**

• Registration for user

**Step 1**: $UV_i$ chooses his/her identity and pseudo-identity pair $(I_i, PI_i)$, password $PA_i$ and a random number $r_i$. $HP_i = h(PA_i || r_i)$ is computed and $(I_i, PI_i)$ is sent to $CS$ through the secure channel by the user.

**Step 2**: $CS$ sees if $I_i$ is valid and will terminate the registration process if the identity is invalid. If it is valid, then $CS$ computes $C_1^* = h(PI_i || I_{cs} || k)$ and $C_2^* = h(I_i || k)$. And the identity $I_i$ is stored in database and $(C_1^*, C_2^*, I_{cs})$ is sent to $UV_i$ through the secure channel by $CS$.

**Step 3**: The user calculates $C_1 = C_1^* \oplus HP_i$, $C_2 = C_2^* \oplus h(I_i || HP_i)$ and $C_3 = r_i \oplus h(I_i || PA_i)$ and stores $(C_1, C_2, C_3, PI_i, I_{cs})$ in his smart card.

**Table 2. Notations in Zhou et al.'s scheme.**

| Notation | Description |
|---|---|
| $I_{cs}, k$ | $CS$'s identity and secret key |
| $SV_j, SI_j, PSI_j$ | The $j$th cloud server, its identity and pseudo-identity |
| $UV_i, I_i, PI_i, PA_i$ | The $i$th user, his/her identity, pseudo-identity, password |
| $AV$ | The attacker |
| $h(\cdot)$ | Hash function |
| $SK_u, SK_s, SK_{cs}$ | $UV_i, SV_j$ and $CS$'s session keys |
| $M_1, M_2, M_3, M_4$ | Messages for the authentication |
| $\|$ | Concatenation operator |
| $\oplus$ | XOR operator |

- Registration for cloud server

**Step 1**: The server $SV_j$'s identity and pseudo-identity pair ($SI_j$, $PSI_j$) is sent to $CS$ through a secure channel by the server.

**Step 2**: $CS$ calculates $B_1 = h(PSI_j||I_{cs}||k)$ and $B_2 = h(SI_j||k)$, stores $S_{ij}$ and ($B_1$, $B_2$, $I_{cs}$) is sent to the server through the secure way.

**Step 3**: ($B_1$, $B_2$, $SI_j$, $PSI_j$, $I_{cs}$) is stored in $SV_j$'s database.

**Authentication phase**

**Step 1**: The user $UV_i$ inserts his/her smart card into the reader and inputs ($I_i$, $PA_i$). And the smart card chooses a nonce $b_u$ and a new pseudo-identity $PI_i^{new}$ and calculates $r_i = C_3 \oplus h(I_i || PA_i)$, $HP_i = h(PA_i|| r_i)$, $C_1^* = C_1 \oplus HP_i$, $C_2^* = C_2 \oplus h(I_i || HP_i)$, $D_1 = C_1^* \oplus b_u$, $D_2 = h(b_u||PI_i||I_{cs}) \oplus I_i$, $D_3 = C_2^* \oplus h(I_i ||HP_i) \oplus PI_i^{new} \oplus h(b_u || I_i)$, $D_4 = h(I_i||PI_i||PI_i^{new}||b_u||D_3)$. Then the user sends the message $M_1 = \{PI_i, D_1, D_2, D_3, D_4\}$ to the nearest cloud server $SV_j$.

**Step 2**: The server chooses $PSI_j^{new}$ and a nonce $b_s$, calculates $D_5 = B_1 \oplus b_s$, $D_6 = h(b_s||PSI_j||I_{cs}) \oplus SI_j$, $D_7 = B_2 \oplus PSI_j^{new} \oplus h(b_s || SI_j)$, $D_8 = h(SI_j||PSI_j||PSI_j^{new}||b_s||D_7)$. And $SV_j$ sends the message $M_2 = \{PI_i, D_1, D_2, D_3, D_4, PSI_j, D_5, D_6, D_7, D_8\}$ to $CS$ through the secure channel.

**Step 3**: $b_u = D_1 \oplus h(PI_i||I_{cs}||k)$, $I_i = D_2 \oplus h(b_u||PI_i||I_{cs})$, $PI_i^{new} = D_3 \oplus h(I_i || k) \oplus h(b_u || I_i)$ is computed by $CS$ and $CS$ sees if $I_i$ is valid and sees if $D_4 = h(I_i||PI_i||PI_i^{new}||b_u||D_3)$. If so, $CS$ computes $b_s = D_5 \oplus h(PSI_j||I_{cs}||k)$, $SI_j = D_6 \oplus h(b_s||PSI_j||I_{cs})$, $PSI_j^{new} = D_7 \oplus h(SI_j ||k) \oplus h(b_s || SI_j)$ and sees if $S_{ij}$ is valid and sees if $D_8 = h(SI_j||PSI_j||PSI_j^{new}||b_s||D_7)$. $CS$ will stop the session if any verification is not right. Or else, $CS$ chooses a nonce $b_{cs}$ and computes $SK_{cs} = h(b_u \oplus b_s \oplus b_{cs})$, $D_9 = h(PSI_j^{new}||I_{cs}||k) \oplus h(b_s || PSI_j^{new})$, $D_{10} = h(PSI_j^{new}||b_s||PSI_j) \oplus (b_u \oplus b_{cs})$, $D_{11} = h(SK_{cs}||D_9||D_{10} || h(SI_j || k))$, $D_{12} = h(PI_i^{new}||I_{cs}||k) \oplus h(b_u|| PI_i^{new})$, $D_{13} = h(PI_i^{new}||b_u||PI_i) \oplus (b_s \oplus b_{cs})$, $D_{14} = h(SK_{cs}||D_{12}||D_{13}|| h(I_i || k))$. And $CS$ sends the message $M_3 = \{D_9, D_{10}, D_{11}, D_{12}, D_{13}, D_{14}\}$ to $SV_j$.

**Step 4**: $(b_u \oplus b_{cs}) = D_{10} \oplus h(PSI_j^{new}||b_s||PSI_j)$ and $SK_s = h(b_s \oplus b_u \oplus b_{cs})$ are computed by $SV_j$. And the server sees if $D_{11} = h(SK_s||D_9||D_{10} || B_2)$ is true. If it's true, $SV_j$ computes $B_1^{new} = D_9 \oplus h(b_s ||PSI_j^{new})$ and replaces ($B_1$, $PSI_j$) with ($B_1^{new}$, $PSI_j^{new}$). In the end, the server sends the message $M_4 = \{D_{12}, D_{13}, D_{14}\}$ to $UV_i$.

**Step 5**: After receiving $M_4$, the smart card calculates $(b_s \oplus b_{cs}) = D_{13} \oplus h(PI_i^{new}||b_u||PI_i)$, $SK_u = h(b_u \oplus b_s \oplus b_{cs})$ and sees if $D_{14}? = h(SK_u||D_{12}||D_{13} || C_2^*)$ is true. If passed, the smart card calculates $C_1^{new} = D_{12} \oplus h(b_u || PI_i^{new}) \oplus HP_i$ and replaces ($C_1$, $PI_i$) with ($C_1^{new}$, $PI_i^{new}$).

**Password update phase**

**Step 1**: When the user $UV_i$ wants to change his/her password, he/she inserts his/her smart card into the reader and inputs ($I_i$, $PA_i$). And the smart card chooses a nonce $b_u$ and a new pseudo-identity $PI_i^{new}$ and calculates $r_i = C_3 \oplus h(I_i || PA_i)$, $HP_i = h(PA_i || r_i)$, $C_1^* = C_1 \oplus HP_i$, $C_2^* = C_2 \oplus h(I_i || HP_i)$, $D_1 = C_1^* \oplus b_u$, $D_2 = h(b_u||PI_i||I_{cs}) \oplus I_i$, $D_3 = C_2^* \oplus h(I_i || HP_i) \oplus PI_i^{new} \oplus h(b_u || I_i)$ and $D_4 = h(I_i||PI_i||PI_i^{new}||b_u||D_3)$. Then the user sends the message $M_5 = \{PI_i, D_1, D_2, D_3, D_4\}$ with a password change request to $CS$.

**Step 2**: $CS$ calculates $b_u$, $I_i$, $P_i$ and sees if $I_i$ and $D_4$ are correct. If so, $D_{12}$ and $D_{15} = h(I_i||PI_i||PI_i^{new}||b_u||D_{12})$ is computed by $CS$. In the end, $M_6 = \{D_{12}, D_{15}\}$ with a permission is sent to $UV_i$.

**Step 3**: The smart card sees if $D_{15} = h(I_i||PI_i||PI_i^{new}||b_u||D_{12})$ is true. If so, it prompts $UV_i$ to enter a new password $PA_i^{new}$ and calculates $HP_i^{new} = h(PA_i^{new} || r_i)$, $C_1^{new2} = D_{12} \oplus h(b_u ||$

$PI_i^{new}) \oplus HP_i^{new}$, $C_2^{new} = C_2^* \oplus h(I_i \| HP_i^{new})$ and $C_3^{new} = r_i \oplus h(I_i \| PA_i^{new})$ and replaces $(C_1, C_2, C_3, PI_i)$ with $(C_1^{new2}, C_2^{new}, C_3^{new}, PI_i^{new})$.

**3.2.2 Cryptanalysis of Zhou et al.'s scheme.** *No providing mutual authentication.* Zhou et al.'s scheme doesn't provide the mutual authentication between the user and the server. At first, in the step 1 of authentication phase, the message $M_1 = \{PI_i, D_1, D_2, D_3, D_4\}$ which is sent to the server by the user doesn't include the information concerned with the server $SV_j$. And in the step 2 of authentication phase, after receiving the message $M_1$, $SV_j$ directly computes $D_5$, $D_6$, $D_7$ and $D_8$ without verifying the user which sends $M_1$, and then sends the message $M_2 = \{PI_i, D_1, D_2, D_3, D_4, PSI_j, D_5, D_6, D_7, D_8\}$ to the registration center $CS$. Also in the step 3, $CS$ authenticates $SV_j$ by checking if $D_8 = h(SI_j \| PSI_j \| PSI_j^{new} \| b_s \| D_7)$ and authenticates $UV_i$ by checking if $D_4 = h(I_i \| PI_i \| PI_i^{new} \| b_u \| D_3)$. In the step 4, when the server receives the message $M_3 = \{D_9, D_{10}, D_{11}, D_{12}, D_{13}, D_{14}\}$, the server authenticates $CS$ by checking if $D_{11} = h(SK_s \| D_9 \| D_{10} \| B_2)$. But the message $M_3$ doesn't contain the information that the server can authenticate the user $UV_i$. In this step, the information concerned with $UV_i$ is only $(b_u \oplus b_{cs})$ but using it, the server cannot authenticate the user $UV_i$. So in this step, the server can authenticate $CS$, but it cannot authenticate the user $UV_i$. Continuously in the step5, $UV_i$ receives the message $M_4 = \{D_{12}, D_{13}, D_{14}\}$ from the server and authenticates $CS$ by checking if $D_{14} = h(SK_u \| D_{12} \| D_{13} \| C_2^*)$. But the information that the user can authenticate the server $SV_j$ isn't included in the message $M_4$. In this step, the information concerned with $SV_j$ is only $(b_s \oplus b_{cs})$ but using it, the user cannot authenticate $SV_j$. So in this step, the user can authenticate $CS$, but cannot authenticate the server $SV_j$.

In conclusion, Zhou et al.'s scheme doesn't provide the mutual authentication between the user and the server.

According to the random oracle model [46], the attacker can intercept and steal the message $M_1$ which is sent to the server $SV_j$ by $UV_i$ and he can send it to another server $SV_m$ which is not the server $SV_j$. Then $SV_m$ cannot know that $UV_i$ sent $M_1$ to itself because the mutual authentication isn't provided between them. And the server $SV_m$ sends the message $M_4$ to the attacker after passing from the step2 to the step 4. In that case, the attacker sends back this message to the user $UV_i$ and the user $UV_i$ regards the session key which he computes in the end as the session key between himself and the server $SV_j$ because he doesn't know that he has communicated with the server $SV_m$ until then.

Like this, in this scheme, the user doesn't know which server he is communicating with, so even if the attacker sends his message to the other server, he will never recognize it.

*Replay attack.* In the step2 of the authentication phase, after receiving the message $M_1$ from the user, the server doesn't check if the message $M_1$ was replayed. And the server computes $D_5$, $D_6$, $D_7$, $D_8$ and sends the message $M_2$ to $CS$. And in the step3, the message $M_2$ which the server sent to $CS$ doesn't contain any information which $CS$ can check if $M_1$ was replayed in the step1 such as a time stamp or the random number which $CS$ generated and sent to the user. In this step, $CS$ only knows the random number $b_u$ generated by the user. So $CS$ also doesn't recognise the replay attack. Thus, if the attacker steals the message $M_1$ and retransmits it to the server $SV_j$, the server will not recognize this attack and will keep computing. After receiving $M_1$, the server will send $M_2$ to $CS$ and $CS$ also will not recognize the replay attack and will transmit $M_3$ to the server. Finally, the server will send $M_4$ to the attacker. Like this way, the attacker can pass the step2, 3 and 4 very easily. As a result, this scheme is vulnerable to the replay attack.

*No key control.* In Zhou et al.'s scheme, the session key is computed as follow: $SK_s = h(b_s \oplus b_u \oplus b_{cs})$. Here, $b_s$ is the random number generated by the server, $b_u$ is the random number generated by the user and $b_{cs}$ is the random number generated by $CS$. In the step3 of the

authentication phase, $CS$ gets $b_u$ and $b_s$ by computing $b_u = D_1 \oplus h (PI_i||I_{cs}||k)$ and $b_s = D_5 \oplus h (PSI_j||I_{cs}||k)$. And $CS$ generates the random number $b_{CS}$ and computes the session key $SK_{cs} = h (b_u \oplus b_s \oplus b_{cs})$. Like this, $CS$ knows all of the three random numbers so the session key between the user and the server publishes to $CS$. As we know, the session key is the secret value which only two session entities must have because this is the key for the session between the user and the server. But in this scheme, $CS$ also gets the session key between the user and the server. Thus, this scheme doesn't provide no key control property.

*Re-registration with the original identity.* In the user registration process of this scheme, the secret value concerned with the user identity is $C_2{}^* = h(I_i || k)$. In the case that $CS$'s secret key $k$ is known to the attacker, the user has to update the value of $C_2{}^*$ concerned with his identity by reregistration in $CS$. So the attacker cannot guess the next round's secret value. But in this scheme, the user cannot register again with his original identity $I_i$ and he can no longer use this identity because the secret $C_2{}^*$ doesn't include any nonce.

If a random number is included in the computing of $C_2{}^*$, the user can use the original identity because he can update $C_2{}^*$ by generating the new random number.

*Inefficiency in the verification of wrong password.* In 2006, Tsai et al. [47] pointed out that the ideal password-based scheme should detect typo error quickly without the communication with the home server. But in Zhou et al.'s scheme, if the attack inputs wrong password, it can not be quickly detected by the smart card. In the step 1 of the authentication phase, the smart card gets $r_i = C_3 \oplus h(I_i || PA_i)$ when the user enters his identity and password. But in this step, the smart card doesn't check if this $r_i$ is the same with the random number generated by the user in the registration phase, keeps computing and sends the message $M_1$ to the server. In the step2, there is also no verification process of the password and in this step the server sends the message $M_2$ to $CS$. Only then in the step3, $CS$ can recognise the wrong password by checking the value of $D_4$.

Therefore, if the attacker inputs the wrong password in the step1, all the values computed in this step will be wrong. But the authentication process keeps going passing the step2 and step3, only then in the step3 these errors are detected.

# 4. Proposed scheme

In this section we describe an improved authentication key agreement protocol using pseudo-identity that overcomes the limitations of the Zhou et al.'s scheme. The proposed scheme consists of four steps: registration phase, authentication, session key exchange phase, password change phase and user revocation, reregistration phase. The notations in Table 3 are used to describe the proposed scheme in our work.

## 4.1 Registration phase

**4.1.1 User registration phase.**　All users who want to exchange session keys using the proposed scheme must register on $CS$.

Fig 1 shows the user registration process.

**Step 1**: The user $UR_i$ selects $UID_i$, $PUID_i$, $PW_i$ and inputs $BIO_i$ in the smart card. Then the smart card extracts $(R_i, P_i)$ from $Gen(BIO_i) \rightarrow (R_i, P_i)$, computes $VD_i = h(PW_i||R_i||UID_i)$ and sends $(UID_i, PUID_i)$ to $R$ via a secure channel.

**Step 2**: The registration center $R$ generates a random number $RU_c$, computes $UD_1{}^* = h (PUID_i||RID||x_U)$, $UD_2{}^* = h(UID_i||x_U||RU_{cs})$ and stores $UID_i$, $RU_{cs}$ in its database. And then $R$ sends $(UD_1{}^*, UD_2{}^*, RID)$ to $UR_i$ via a secure channel.

**Table 3. Notation used in proposed scheme.**

| Notation | Description |
|---|---|
| $R$ | Registration center |
| $UR_i, SR_j$ | The $i$th user, $j$th server |
| $x_U$ | Secret key which $R$ shares with $UR_i$ |
| $x_S$ | Secret key which $R$ shares with $SR_j$ |
| $SC_i$ | Smart card of $UR_i$ |
| $UID_i$ | Identity of $UR_i$ |
| $PUID_i$ | Pseudo-identity of $UR_i$ |
| $RID$ | Identity of $R$ |
| $SID_j$ | Identity of $SR_j$ |
| $PSID_j$ | Pseudo-identity of $SR_j$ |
| $PW_i$ | Password of $UR_i$ |
| $BIO_i$ | Biometric of $UR_i$ |
| $SK$ | Session key for $UR_i$ and $SR_j$ |
| $T_n(\alpha)$ | Chebyshev chaotic map |
| $h(\cdot)$ | One-way hash function |
| $\parallel$ | Concatenation operator |
| $\oplus$ | XOR operator |

**Step 3**: The user computes $UD_1 = UD_1{}^*\oplus VD_i$, $UD_2 = UD_2{}^*\oplus VD_i$, $UD_3 = h(VD_i\|UD_1{}^*\|UD_2{}^*)$ and stores $(UD_1, UD_2, UD_3, PUID_i, RID, P_i)$ in his smart card.

**4.1.2 Server registration phase.** Fig 2 shows the server registration process.

**Step 1**: The server $SR_j$ first selects $SID_j$, $PSID_j$ and sends $(SID_j, PSID_j)$ to $R$ via a secure channel.

**Step 2**: $R$ computes $SD_1 = h(PSID_j\|RID\|x_S)$, $SD_2 = h(SID_j\|x_S\|RS_{cs})$ and stores $SID_j$, $RS_{cs}$ in the database. And then $R$ sends $(SD_1, SD_2, RID)$ to the server.

**Step 3**: The server $SR_j$ stores $(SD_1, SD_2, SID_j, PSID_j, RID)$ in his smart card.

## 4.2 Authentication and session key exchange phase

Fig 3 shows the authentication and session key exchange steps of the proposed scheme.

**Step 1**: The user $UR_i$ inserts his smart card into a card reader and enters $UID_i$, $PW_i$ and $BIO_i{}^*$. The smart card recovers $R_i$ from $Rep(BIO_i{}^*, P_i) \rightarrow R_i$, selects the random numbers $r_U$, $PUID_i{}^{new}$, $rk_U$ and computes $P_U = T_{rkU}(\alpha)$ mod $p$, $UD_1{}^* = UD_1\oplus h(PW_i\|R_i\|UID_i)$, $UD_2{}^* = UD_2\oplus h(PW_i\|R_i\|UID_i)$. And then $UR_i$ computes $UD_3{}' = h(h(PW_i\|R_i\|UID_i)\|UD_1{}^*\|UD_2{}^*)$ and checks if $UD_3{}' = UD_3$. If it's false, this phase will be stopped. If so, the smart card calculates $E_1 = UD_1{}^*\oplus r_U$, $E_2 = h(r_U\|PUID_i\|RID)\oplus UID_i$, $E_3 = PUID_i{}^{new}\oplus h(r_U\|UID_i)$, $V_{UR} = h(UID_i\|PUID_i\|PUID_i{}^{new}\|r_U\|P_U\|SID_j\|T_1\|UD_2{}^*)$ ($T_1$ is a time stamp.) and sends the message $M_1 = \{PUID_i, E_1, E_2, E_3, V_{UR}, P_U, T_1\}$ to the server $SR_j$.

**Step 2**: After receiving the message $M_1$, the server $SR_j$ computes $\Delta T = T_1 - T_2$ that is the difference between $T_1$ and $T_2$ ($T_2$ is the current time on $SR_j$). If the difference $\Delta T$ is greater than $\Delta T_{define}$ that is the defined time interval, the server will stop the authentication phase. Else the server selects $PSID_j{}^{new}$, $r_S$, $rk_S$ and calculates $P_S = T_{rkS}(\alpha)$ mod $p$, $SK = T_{rkS}(P_U) = T_{rkS,rkU}(\alpha)$ mod

| **User ($UR_i$)** | **Registration Center ($R$)** |
|---|---|
| Selects $UID_i$, $PUID_i$, $PW_i$ and inputs $BIO_i$ | |
| extracts $(R_i, P_i)$ from $Gen(BIO_i) \rightarrow (R_i, P_i)$ | |
| $VD_i = h(PW_i\|\|R_i\|\|UID_i)$ | |

$$\xrightarrow[\text{via a secure channel}]{(UID_i, PUID_i)}$$

Selects $RU_{cs}$

$UD_1^* = h(PUID_i\|\|RID\|\|x_U)$

$UD_2^* = h(UID_i\|\|x_U\|\|RU_{cs})$

Stores $UID_i, RU_{cs}$ in the database

$$\xleftarrow[\text{via a secure channel}]{(UD_1^*, UD_2^*, RID)}$$

$UD_1 = UD_1^* \oplus VD_i$

$UD_2 = UD_2^* \oplus VD_i$

$UD_3 = h(VD_i\|\|UD_1^*\|\|UD_2^*)$

Stores $(UD_1, UD_2, UD_3, PUID_i, RID, P_i)$ in $SC$

**Fig 1. User registration phase in the proposed scheme.**

$p$, $E_4 = SD_1 \oplus r_S$, $E_5 = h(r_S\|\|PSID_j\|\|RID) \oplus SID_j$, $E_6 = PSID_j^{new} \oplus h(r_S\|\|SID_j)$, $V_{SU} = h(SK\|\|SID_j\|\|RID)$, $V_{SR} = h(SID_j\|\|PSID_j\|\| PSID_j^{new}\|\|r_S\|\|P_U\|\|P_S\|\|T_3\|\|V_{SU}\|\| SD_2)$ ($T_3$ is a time stamp.). And $SR_j$ transmits the message $M_2 = \{PUID_i, E_1, E_2, E_3, V_{UR}, E_4, E_5, E_6, V_{SR}, T_1, T_3, P_S, P_U, V_{SU}\}$ to the registration center $R$.

| **Server ($SR_j$)** | **Registration Center ($R$)** |
|---|---|
| Selects $SID_j$ and $PSID_j$ | |

$$\xrightarrow[\text{via a secure channel}]{(SID_i,\ PSID_i)}$$

$SD_1 = h(PSID_j\|\|RID\|\|x_S)$

$SD_2 = h(SID_j\|\|x_S\|\|RS_{cs})$

Stores $SID_j$ and $RS_{cs}$ in the database

$$\xleftarrow[\text{via a secure channel}]{(SD_1,\ SD_2,\ RID)}$$

Stores $(SD_1, SD_2, SID_j, PSID_j, RID)$ in $SC_j$

**Fig 2. Server registration phase in the proposed scheme.**

| User ($UR_i$/$SC_i$) | Server ($SR_j$) | Registration Center ($R$) |
| --- | --- | --- |

Inputs $UID_i, PW_i, BIO_i^*$ in $SC_i$, $Rep(BIO_i^*, P_i) \to R_i$

$SC_i$ selects $r_U, PUID_i^{new}$ and $rk_U$

$P_U = T_{rk_i}(\alpha) \bmod p$

$UD_1^* = UD_1 \oplus h(PW_i||R_i||UID_i)$

$UD_2^* = UD_2 \oplus h(PW_i||R_i||UID_i)$

$UD_3' = h(h(PW_i||R_i||UID_i)||UD_1^*||UD_2^*)$

Checks if $UD_3' = UD_3$

$E_1 = UD_1^* \oplus r_U$, $E_2 = h(r_U||PUID_i||RID) \oplus UID_i$

$E_3 = PUID_i^{new} \oplus h(r_U||UID_i)$

$V_{UR} = h(UID_i||PUID_i||PUID_i^{new}||r_U||P_U||SID_j||T_1||UD_2^*)$

$M_1 = \{PUID_i, E_1, E_2, E_3, V_{UR}, P_U, T_1\}$

Checks if $\Delta T = T_1 - T_2 < \Delta T_{define}$

Selects $PSID_j^{new}, r_S$ and $rk_S$

$P_S = T_{rk_i}(\alpha) \bmod p$, $SK = T_{rk_i}(P_U) = T_{rk,rk_i}(\alpha) \bmod p$

$E_4 = SD_1 \oplus r_S$, $E_5 = h(r_S||PSID_j||RID) \oplus SID_j$

$E_6 = PSID_j^{new} \oplus h(r_S||SID_j)$, $V_{SU} = h(SK||SID_j||RID)$

$V_{SR} = h(SID_j||PSID_j||PSID_j^{new}||r_S||P_U||P_S||T_3||V_{SU}||SD_2)$

$M_2 = \{PUID_i, E_1, E_2, E_3, V_{UR}, E_4, E_5, E_6, V_{SR}, T_1, T_3, P_S, P_U, V_{SU}\}$

Checks if $\Delta T^* = T_3 - T_4 < \Delta T^*_{define}$

$r_U = E_1 \oplus h(PUID_i||RID||x_U)$  $UID_i = E_2 \oplus h(r_U||PUID_i||RID)$

$UD_2^* = h(UID_i||x_U||RU_{cs})$, $PUID_i^{new} = E_3 \oplus h(r_U||UID_i)$

$r_S = E_4 \oplus h(PSID_j||RID||x_S)$, $SID_j = E_5 \oplus h(r_S||PSID_j||RID)$

$SD_2 = h(SID_j||x_S||RS_{cs})$, $PSID_j^{new} = E_6 \oplus h(r_S||SID_j)$

Checks if $UID_i$ is valid

Checks if $V_{UR}? = h(UID_i||PUID_i||PUID_i^{new}||r_U||P_U||SID_j||T_1||UD_2^*)$

Checks if $SID_j$ is valid

Checks if $V_{SR}? = h(SID_j||PSID_j||PSID_j^{new}||r_S||P_U||P_S||T_3||V_{SU}||SD_2)$

$E_7 = h(PSID_j^{new}||RID||x_S) \oplus h(r_S||PSID_j)$

$V_{RS} = h(SD_2^*||r_S||P_U)$

$E_8 = h(PUID_i^{new}||RID||x_U) \oplus h(r_U||PUID_i)$

$V_{RU} = h(UD_2^*||r_U||P_S)$

$M_3 = \{E_7, E_8, V_{RU}, V_{RS}, P_S, V_{SU}\}$

Checks if $V_{RU}? = h(UD_2^*||r_U||P_S)$

$SK = T_{rk_i}(P_S) = T_{rk,rk_i}(\alpha) \bmod p$

Checks if $V_{SU}? = h(SK||SID_j||RID)$

$UD_1^{*new} = E_8 \oplus h(r_U||PUID_i)$, $UD_1^{new} = UD_1^{*new} \oplus h(PW_i||R_i||UID_i)$

Replaces $(UD_1^{new}, PUID_i^{new})$ with $(UD_1, PUID_i)$

$V_{US} = h(SK||RID)$

$M_4 = \{E_7, V_{RS}, V_{US}\}$

Checks if $V_{RS}? = h(SD_2||r_S||P_U)$, Checks if $V_{US}? = h(SK||RID)$

$SD_1^{new} = E_7 \oplus h(r_S||PSID_j)$, replaces $(SD_1^{new}, PSID_j^{new})$ with $(SD_1, PSID_j)$

$SK = T_{rk,rk_i}(\alpha) \bmod p$

**Fig 3. Authentication and session key exchange phase of the proposed scheme.**

**Step 3**: And $R$ computes $\Delta T^* = T_3 - T_4$ that is the difference between $T_3$ and $T_4$ ($T_4$ is the current time on $R$). If $\Delta T^*$ is greater than $\Delta T^*_{define}$ that is the defined time interval, $R$ will stop this phase. Else $R$ computes the following data: $r_U = E_1 \oplus h(PUID_i||RID||x_U)$, $UID_i = E_2 \oplus h(r_U|| PUID_i||RID)$, $UD_2^{*'} = h(UID_i||x_U||RU_{cs})$, $PUID_i^{new} = E_3 \oplus h(r_U||UID_i)$, $r_S = E_4 \oplus h(PSID_j|| RID||x_S)$, $SID_j = E_5 \oplus h(r_S ||PSID_j||RID)$, $SD_2^* = h(SID_j||x_S||RS_{cs})$, $PSID_j^{new} = E_6 \oplus h(r_S||SID_j)$. And then $R$ checks if the identity $UID_i$ is valid. If so, $R$ computes $V_{UR}' = h(UID_i||PUID_i||$

$PUID_i^{new}||r_U||P_U||SID_j||T_1||UD_2^*$) and checks if $V_{UR}' = V_{UR}$. If not so, the session will be terminated. Else it is checked if $SID_j$ is valid, computes $V_{SR} = h(SID_j||PSID_j|| PSID_j^{new}||r_S||P_U|| P_S||T_3||V_{SU}|| SD_2)$ and checks if $V_{SR}' = V_{SR}$. If so, $E_7 = h(PSID_j^{new}||RID||x_S) \oplus h(r_S||PSID_j)$, $V_{RS} = h(SD_2^*||r_S||P_U)$, $E_8 = h(PUID_i^{new}||RID||x_U) \oplus h(r_U||PUID_i)$, $V_{RU} = h(UD_2^{*}||r_U||P_S)$ are computed and $R$ sends the message $M_3 = \{E_7,E_8,V_{RU},V_{RS},P_S,V_{SU}\}$ to $UR_i$.

**Step 4**: Then the user calculates $V_{RU}' = h(UD_2^*||r_U||P_S)$ and checks if $V_{RU}' = V_{RU}$. If so, he calculates $SK' = T_{rkU}(P_S) = T_{rkSrkU}(\alpha) \bmod p$, $V_{SU}' = h(SK'||SID_j||RID)$ and checks if $V_{SU}' = V_{SU}$. If not so, the session will be stopped. Else he keeps $SK'$ as the session key $SK$. And the user computes $UD_1^{*new} = E_8 \oplus h(r_U||PUID_i)$, $UD_1^{new} = UD_1^{*new} \oplus h(PW_i||BIO_i||UID_i)$ and replaces $(UD_1,PUID_i)$ with $(UD_1^{new},PUID_i^{new})$. Also he calculates $V_{US} = h(SK||RID)$ and transmits the message $M_4 = \{E_7, V_{RS}, V_{US}\}$ to the server $SR_j$.

**Step 5**: The server $SR_j$ computes $V_{RS}' = h(SD_2||r_S||P_U)$ and checks if $V_{RS}' = V_{RS}$. If so, $SR_j$ computes $V_{US}' = h(SK||RID)$ and checks if $V_{US} = V_{US}'$. If not so, the session will be terminated. Else the server regards $SK$ as the session key for itself and $UR_i$, computes $SD_1^{new} = E_7 \oplus h(r_S|| PSID_j)$ and replaces $(SD_1, PSID_j)$ with $(SD_1^{new}, PSID_j^{new})$.

## 4.3 Password change phase

**Step 1**: When the user $UR_i$ wants to change his password, he first inserts his smart card into a reader and inputs $UID_i$, $PW_i$ and $BIO_i$. Then the smart card extracts $(R_i, P_i)$ from $Gen(BIO_i) \rightarrow (R_i, P_i)$, selects $r_U$, $PUID_i^{new}$, $rk_U$, computes $P_U = T_{rkU}(\alpha) \bmod p$, $UD_1^* = UD_1 \oplus h(PW_i||R_i||UID_i)$, $UD_2^* = UD_2 \oplus h(PW_i||R_i||UID_i)$, $UD_3' = (h(PW_i||R_i||UID_i)||UD_1^*||UD_2^*)$ and checks if $UD_3' = UD_3$. If so, the smart card calculates $E_1 = UD_1 \oplus r_U$, $E_2 = h(r_U||PUID_i||RID) \oplus UID_i$ and $E_3 = UD_2^* \oplus PUID_i^{new} \oplus h(r_U||UID_i)$, $V_{UR} = h(UID_i||PUID_i||PUID_i^{new}||r_U||P_U|| SID_j||T_5)$ ($T_5$ is a time stamp.), and sends the message $M_5 = \{PUID_i,E_1,E_2,E_3,V_{UR}, P_U,T_5\}$ with a password change request to $R$.

**Step 2**: After receiving the message with a request, $R$ calculates $\Delta T^{**} = T_5 - T_6$ that is a difference between $T_5$ and $T_6$ ($T_6$ is the current time on $R$). If $\Delta T^{**}$ is greater than a defined time interval, $R$ will terminate this phase. Else $R$ computes $r_U = E_1 \oplus h(PUID_i||RID||x_U)$, $UID_i = E_2 \oplus h(r_U||PUID_i||RID)$, $UD_2^{*'} = h(UID_i||x_U||RU_{cs})$, $PUID_i^{new} = E_3 \oplus UD_2^{*'} \oplus h(r_U||UID_i)$. And $R$ checks if $UID_i$ is valid, computes $V_{UR}' = h(UID_i||PUID_i||PUID_i^{new}||r_U||P_U||SID_j||T_1)$ and checks if $V_{UR}' = V_{UR}$. If so, $E_8 = h(PUID_i^{new}||RID||x_U) \oplus h(r_U||PUID_i)$, $E_9 = h(UID_i||PUID_i|| PUID_i^{new}||r_U||D_8)$ are computed and the message $M_6 = \{E_8,E_9\}$ is sent to the user $UR_i$.

**Step 3**: And the smart card of $UR_i$ calculates $E_9' = h(UID_i||PUID_i||PUID_i^{new}||r_U||E_8)$ and checks if $E_9' = E_9$. If so, $UR_i$ enters the new password $PW_i^{new}$. Then the smart card calculates $VD_i^{new} = h(PW_i^{new}||R_i||UID_i)$, $UD_1^{new2} = UD_1^{new} \oplus VD_i^{new}$, $UD_2^{new} = UD_2^* \oplus VD_i^{new}$, $UD_3^{new} = h(VD_i^{new}||UD_1^{new}||UD_2^*)$ and replaces $(UD_1,UD_2,UD_3)$ with $(UD_1^{new2},UD_2^{new},UD_3^{new})$.

## 4.4 User revocation and re-registration phase

**4.4.1 User revocation phase.**   If the user $UR_i$ wants to revocate his data on the registration center, he needs to send his identity $UID_i$ with a revocation request to $R$ via a secure channel. Then $R$ checks if $UID_i$ was included in its database and if so, $R$ will remove all data concerned with the identity $UID_i$ in the database. And a revocation response is sent to the user $UR_i$ via a secure channel.

**4.4.2 Reregistration phase.**   If the user $UR_i$ wants to register again, he has to choose his new identity, new pseudo-identity, new password, new biometric and to pass the steps in the

user registration phase as they are. In the precedent schemes, the user's secret key $UD_1^* = h(UID_i||x_U)$ consists of only the user's identity and the server's secret value. Thus, if the user's secret key is published, the user must choose a new identity to register again. But in our scheme, the user's secret key $UD_1^* = h(UID_i||x_U||RU_{cs})$ consists of the user's identity, the server's secret value and a nonce. The user's secret key is updated without the identity change because the registration center $R$ generates a new nonce $RU_{cs}$ every registration phase. Therefore, the user can register again on $R$ without change of his identity. To register again on $R$ with an original identity, the user has to transmit his original identity, pseudo-identity, password and biometric to $R$. And then he has to pass the steps in the user registration phase as they are.

## 5. Security analysis of the proposed scheme

In this section, we analyse the security properties of the proposed scheme. First, we prove the validation of the session key between the user and server by using BAN logic [48]. Next, we simulate the proposed scheme for the formal security analysis by using AVISPA (Automated validation of internet security protocol and application) tool [49]. Last, we demonstrate the proposed scheme can resist various kinds of attacks.

### 5.1 Authentication proof based on BAN logic

**Notations and rules.** We define $P$ and $Q$ as the specific participators, $S$ is the trusted server, and $X$ is the formula (statement). Some notations and rules of BAN logic are as follows [48].

$P|\equiv X$: $P$ believes $X$.

$P \triangleleft X$: $P$ sees $X$.

$P|\sim X$: $P$ once said $X$.

$P|\Rightarrow X$: $P$ has jurisdiction over $X$.

$\#(X)$: $X$ is fresh.

$P \xleftrightarrow{K} Q$: $K$ is a shared secret key between $P$ and $Q$.

$\{X\}_K$: Formula $X$ is encrypted under the key $K$.

$<X>_Y$: $X$ combined with the formula $Y$.

$R_1 : \frac{P|\equiv Q \xleftrightarrow{K} P,\ P \triangleleft \{X\}_K}{P|\equiv Q|\sim X}$ (Message-meaning rule): if $P$ believes that the key $K$ is shared with $Q$ and receives a message containing $X$ encrypted under $K$, then $P$ believes that $Q$ once said $X$.

$R_2 : \frac{P|\equiv \#(X),\ P|\equiv Q|\sim X}{P|\equiv Q|\equiv X}$ (Nonce-verification rule): if $P$ believes $X$ is fresh and $Q$ once said $X$, $P$ believes $Q$ believes $X$.

$R_3 : \frac{P|\equiv Q|\Rightarrow X,\ P|\equiv Q|\equiv X}{P|\equiv X}$ (Jurisdiction rule): if $P$ believes that $Q$ had jurisdiction right to $X$ and believes $Q$ believes $X$, $P$ believes $X$.

$R_4 : \frac{P|\equiv \#(X)}{P|\equiv \#(X,Y)}$ (Freshness rule): If $X$ is a part of message $(X, Y)$ and $X$ is fresh, message $(X, Y)$ is also fresh.

$R_5 : \frac{P|\equiv Q|\equiv (X, Y)}{P|\equiv Q|\equiv X}$ (Belief rule 1): If $P$ believes $Q$ believes the message set $(X, Y)$, $P$ also believes $Q$ believes the message $X$.

$R_6$ : $\frac{P|\equiv X,\ P|\equiv Y}{P|\equiv\ (X,Y)}$ (Belief rule 2): If $P$ believes the message $X$ and $Y$, $P$ also believes the message set $(X, Y)$.

$R_7$ : $\frac{P|\equiv Q|\sim H(X),\ P\triangleleft X}{P|\equiv\ Q|\sim X}$ (Hash function rule): if $P$ believes that $Q$ once said $H(X)$ and receives $X$, $P$ believes $Q$ once said $X$.

**Goals.** The session key exchange protocol should achieve the following goals:

$Goal_1 : UR| \equiv UR \xleftrightarrow{SK} SR$

$Goal_2 : SR| \equiv UR \xleftrightarrow{SK} SR$

$Goal_3 : UR| \equiv SR| \equiv UR \xleftrightarrow{SK} SR$

$Goal_4 : SR| \equiv UR| \equiv UR \xleftrightarrow{SK} SR$

**Idealize.** We idealize the communication messages of the proposed scheme as follows:

$$M_1 : UR \rightarrow SR : \{PUID, E_1, E_2, E_3, V_{UR} = < H(UID \parallel PUID \parallel PUID^{new} \parallel r_U \parallel P_U \parallel SID \parallel T_1 \parallel$$
$$UR \xleftrightarrow{UD_{2}*} R)>_{UR \xleftrightarrow{UD_{2}*} R}\}$$

$$M_2 : SR \rightarrow R : \{PUID, E_1, E_2, E_3, V_{UR} = < H(UID \parallel PUID \parallel PUID^{new} \parallel r_U \parallel P_U \parallel SID \parallel T_1 \parallel UR \xleftrightarrow{UD_{2}*} R)>_{UR \xleftrightarrow{UD_{2}*} R},$$
$$E_4, E_5, E_6, V_{SR} = < H(SID \parallel PSID \parallel PSID^{new} \parallel r_S \parallel P_U \parallel P_S \parallel T_3 \parallel V_{SU} \parallel SR \xleftrightarrow{SD_2} R)>_{SR \xleftrightarrow{SD_2} R},$$
$$T_1, T_3, P_S, P_U, V_{SU} = H(SK \parallel SID \parallel RID)\}$$

$$M_3 : R \rightarrow UR : \{E_7, E_8, V_{RU} = < H(UR \xleftrightarrow{UD_{2}*} R \parallel r_U \parallel P_S)>_{UR \xleftrightarrow{UD_{2}*} R}, V_{RS} = < H(SR \xleftrightarrow{SD_2} R \parallel r_S \parallel P_U)>_{SR \xleftrightarrow{SD_2} R},$$
$$P_S, V_{SU} = H(SK \parallel SID \parallel RID)\}$$

$$M_4 : UR \rightarrow SR : \{E_7, V_{RS} = < H(SR \xleftrightarrow{SD_{2}*} R \parallel r_S \parallel P_U)>_{SR \xleftrightarrow{SD_2} R}, V_{US} = H(SK \parallel RID)\}$$

**Assumptions.** The initial assumptions of the proposed scheme are as follows:

$A_{UR1}$: $UR|\equiv rk_U$

$A_{UR2}$: $UR|\equiv\#(rk_U)$

$A_{UR3}$: $UR|\equiv R|\Rightarrow P_S$

$A_{UR4}$ : $UR| \equiv UR \xleftrightarrow{UD_{2}*} R$

$A_{UR5}$: $UR|\equiv r_U$

$A_{UR6}$: $UR|\equiv\#(r_U)$

$A_{SR1}$: $SR|\equiv rk_S$

$A_{SR2}$: $SR|\equiv\#(rk_S)$

$A_{SR3}$: $SR|{\equiv}R|{\Rightarrow}P_U$

$A_{SR4}$ : $SR| \equiv SR \xleftrightarrow{SD_2} R$

$A_{SR5}$: $SR|{\equiv}r_S$

$A_{SR6}$: $SR|{\equiv}\#(r_S)$

$A_{R1}$ : $R| \equiv UR \xleftrightarrow{UD_2*} R$

$A_{R2}$ : $R| \equiv SR \xleftrightarrow{SD_2} R$

**Analysis.** According to $M_3$ and $A_{UR4}$, we apply the hash function rule ($R_7$), we can obtain:

$$S_1 : \frac{UR| \equiv UR \xleftrightarrow{UD_2*} R, \; UR \triangleleft < H(UR \xleftrightarrow{UD_2*} R||r_U||P_S)>_{UR \xleftrightarrow{UD_2*} R}}{UR| \equiv R| \sim H(UR \xleftrightarrow{UD_2*} R||r_U||P_S)},$$

$$\frac{UR| \equiv R| \sim H(UR \xleftrightarrow{UD_2*} R||r_U||P_S), \; UR \triangleleft \{UR \xleftrightarrow{UD_2*} R, r_U, P_S\}}{UR| \equiv R| \sim (UR \xleftrightarrow{UD_2*} R, r_U, P_S)}$$

According to $M_3$ and $A_{UR6}$, we apply the Freshness rule ($R_4$), we can obtain:

$$S_2 : \frac{UR| \equiv \#(r_U)}{UR| \equiv \# H(UR \xleftrightarrow{UD_2*} R||r_U||P_S)},$$

$$\frac{UR| \equiv \# H(UR \xleftrightarrow{UD_2*} R||r_U||P_S)}{UR| \equiv \#(UR \xleftrightarrow{UD_2*} R, r_U, P_S)},$$

According to $S_1$ and $S_2$, we apply the Nonce-verification rule ($R_2$) and Belief rule 1($R_5$), we can obtain:

$$S_3 : \frac{UR| \equiv \#(UR \xleftrightarrow{UD_2*} R, r_U, P_S), \; UR| \equiv R| \sim (UR \xleftrightarrow{UD_2*} R, r_U, P_S)}{UR| \equiv R| \equiv (UR \xleftrightarrow{UD_2*} R, r_U, P_S)}$$

$$\frac{UR| \equiv R| \equiv (UR \xleftrightarrow{UD_2*} R, r_U, P_S)}{UR| \equiv R| \equiv P_S}$$

According to $S_3$ and $A_{UR3}$, we apply the Jurisdiction rule ($R_3$), we can obtain:

$$S_4 : \frac{UR| \equiv R| \Rightarrow P_S, UR| \equiv R| \equiv P_S}{UR| \equiv P_S}$$

According to $S_4$, $A_{UR1}$ and $SK = T_{rkU}(P_S) \bmod p$, we apply the Belief rule 2($R_6$), we can obtain:

$$S_5 : \frac{UR| \equiv rk_U, UR| \equiv P_S}{UR| \equiv UR \xleftrightarrow{SK} SR} : (Goal_1)$$

According to $M_4$ and $A_{SR4}$, we apply the message meaning rule ($R_1$) and the hash function rule ($R_7$), we can obtain:

$$S_6: \frac{SR| \equiv SR \xleftrightarrow{SD_2} R, \ SR\vartriangleleft < H(SR \xleftrightarrow{SD_2} R||r_S||P_U)>_{SR \xleftrightarrow{SD_2} R}}{SR| \equiv R| \sim H(SR \xleftrightarrow{SD_2} R||r_S||P_U)},$$

$$\frac{SR| \equiv R| \sim H(SR \xleftrightarrow{SD_2} R||r_S||P_U), \ SR\vartriangleleft\{SR \xleftrightarrow{SD_2} R, r_S, P_U\}}{SR| \equiv R| \sim (SR \xleftrightarrow{SD_2} R, r_S, P_U)}$$

According to $M_4$ and $A_{SR6}$, we apply the Freshness rule ($R_4$), we can obtain:

$$S_7: \frac{SR| \equiv \#(r_S)}{SR| \equiv \# H(SR \xleftrightarrow{SD_2} R||r_S||P_U)},$$

$$\frac{SR| \equiv \# H(SR \xleftrightarrow{SD_2} R||r_S||P_U)}{SR| \equiv \#(SR \xleftrightarrow{SD_2} R, r_S, P_U)},$$

According to $S_6$ and $S_7$, we apply the Nonce-verification rule ($R_2$) and Belief rule 1($R_5$), we can obtain:

$$S_8: \frac{SR| \equiv \#(SR \xleftrightarrow{SD_2} R, r_S, P_U), SR| \equiv R| \sim (SR \xleftrightarrow{SD_2} R, r_S, P_U)}{SR| \equiv R| \equiv (SR \xleftrightarrow{SD_2} R, r_S, P_U)}$$

$$\frac{SR| \equiv R| \equiv (SR \xleftrightarrow{SD_2} R, r_S, P_U)}{SR| \equiv R| \equiv P_U}$$

According to $A_{SR3}$ and $S_8$, we apply the jurisdiction rule ($R_3$), we can obtain:

$$S_9: \frac{SR| \equiv R| \Rightarrow P_U, SR| \equiv R| \equiv P_U}{SR| \equiv P_U}$$

According to $A_{SR1}$, $S_9$ and $SK = T_{rkS}(P_U) \bmod p$, we apply the Belief rule 2($R_6$), we can obtain:

$$S_{10}: \frac{SR| \equiv rk_S, SR| \equiv P_U}{SR| \equiv UR \xleftrightarrow{SK} SR} : (Goal_2)$$

According to $M_3$ and $S_5$, we apply the message meaning rule ($R_1$) and the hash function rule ($R_7$), we can obtain:

$$S_{11}: \frac{UR| \equiv UR \xleftrightarrow{SK} SR, UR\vartriangleleft < H(UR \xleftrightarrow{SK} SR||SID||RID)>_{UR \xleftrightarrow{SK} SR}}{UR| \equiv SR| \sim H(UR \xleftrightarrow{SK} SR||SID||RID)},$$

$$\frac{UR| \equiv SR| \sim H(UR \xleftrightarrow{SK} SR||SID||RID), \ UR\vartriangleleft < \{UR \xleftrightarrow{SK} SR, SID, RID\}}{UR| \equiv SR| \sim (UR \xleftrightarrow{SK} SR, SID, RID)},$$

According to $A_{UR2}$, $M_3$ and $SK = T_{rkU}(P_S) \bmod p$, we apply the Freshness rule ($R_4$), we can obtain:

$$S_{12}: \quad \frac{UR| \equiv \#(rk_U)}{UR| \equiv \# UR \xleftrightarrow{SK} SR},$$

$$\frac{UR| \equiv \# UR \xleftrightarrow{SK} SR}{UR| \equiv \# H(UR \xleftrightarrow{SK} SR||SID||RID)}$$

$$\frac{UR| \equiv \# H(UR \xleftrightarrow{SK} SR||SID||RID)}{UR| \equiv \#(UR \xleftrightarrow{SK} SR,SID,RID)}$$

According to $S_{11}$ and $S_{12}$, we apply the Nonce-verification rule ($R_2$) and Belief rule 1($R_5$), we can obtain:

$$S_{13}: \quad \frac{UR| \equiv \#(UR \xleftrightarrow{SK} SR,SID,RID), UR| \equiv SR| \sim (UR \xleftrightarrow{SK} SR, SID, RID)}{UR| \equiv SR| \equiv (UR \xleftrightarrow{SK} SR,SID,RID)},$$

$$\frac{UR| \equiv SR| \equiv (UR \xleftrightarrow{SK} SR,SID,RID)}{UR| \equiv SR| \equiv UR \xleftrightarrow{SK} SR} \qquad : (Goal_3)$$

According to $M_4$ and $S_{10}$, we apply the message meaning rule ($R_1$) and the hash function rule ($R_7$), we can obtain:

$$S_{14}: \quad \frac{SR| \equiv UR \xleftrightarrow{SK} SR, SR \vartriangleleft < H(UR \xleftrightarrow{SK} SR||RID)>_{UR \xleftrightarrow{SK} SR}}{SR| \equiv UR| \sim H(UR \xleftrightarrow{SK} SR||RID)},$$

$$\frac{SR| \equiv UR| \sim H(UR \xleftrightarrow{SK} SR||RID), SR \vartriangleleft < \{UR \xleftrightarrow{SK} SR, RID\}}{SR| \equiv UR| \sim (UR \xleftrightarrow{SK} SR, RID)},$$

According to $A_{SR2}$, $M_4$ and $SK = T_{rkS}(P_U) \bmod p$, we apply the Freshness rule ($R_4$), we can obtain:

$$S_{15}: \quad \frac{SR| \equiv \#(rk_S)}{SR| \equiv \# UR \xleftrightarrow{SK} SR},$$

$$\frac{SR| \equiv \# UR \xleftrightarrow{SK} SR}{SR| \equiv \# H(UR \xleftrightarrow{SK} SR||RID)}$$

$$\frac{SR| \equiv \# H(UR \xleftrightarrow{SK} SR||RID)}{SR| \equiv \#(UR \xleftrightarrow{SK} SR,RID)}$$

According to $S_{14}$ and $S_{15}$, we apply the Nonce-verification rule $(R_2)$ and Belief rule 1$(R_5)$, we can obtain:

$$S_{16}: \quad \frac{SR| \equiv \#(UR \xleftrightarrow{SK} SR, RID), SR| \equiv UR| \sim (UR \xleftrightarrow{SK} SR, RID)}{SR| \equiv UR| \equiv (UR \xleftrightarrow{SK} SR, RID)},$$

$$: (Goal_4)$$

$$\frac{SR| \equiv UR| \equiv (UR \xleftrightarrow{SK} SR, RID)}{SR| \equiv UR| \equiv UR \xleftrightarrow{SK} SR}$$

## 5.2 Validation test based on AVISPA

In this section, we simulate the proposed scheme for the formal security analysis using AVISPA, which is widely used to verify the security properties of designed protocol such as resistance against replay attack and man-in-the-middle attack. This tool implements four back-ends: On-the-Fly-Model-Check (OFMC), Constraint Logic based Attack Searcher (CL-AtSe), SAT-based Model-Checker (SATMC) and Three Automata based on Automatic Approximations for the Analysis of Security Protocols (TA4SP). In order to verify the security properties of the protocol using AVISPA, it needs to be specified in HLPSL (High Level Protocol Specification Language), which is a role-based language: basic roles for representing each participant role, and composition roles for representing scenarios of basic roles. Each role is independent from the other, communicating with the other roles by channels [44]. The output format is generated by using one of the four back-ends.

**Specifying the proposed protocol.** In our HLPSL implementation, we define three basic roles for the user $U$, server $S$ and registration center $R$. Figs 4–6 shows the specifications in HLPSL for the role of $U$, $S$ and $R$.

In Figs 7–9, we show the HLPSL implementation for the role of the session, environment and goal.

In our implementation, we verified the following fifteen secrecy goals and six authentication properties.

- secrecy_of sec_rscs: It represents that the nonce $Rscs$ generated by $R$ is kept secret to the registration center $R$ only.

- secrecy_of sec_vd: It represents that user $U$'s private data $VD$ is kept secret to the user $U$ only.

- secrecy_of sec_rucs: It represents that the nonce $Rucs$ generated by $R$ is kept secret to the registration center $R$ only.

- secrecy_of sec_xs: It represents that registration center $R$'s secret key $X_S$ is kept secret to $R$ only.

- secrecy_of sec_b1: It represents that the server $S$'s shared secret key $B_1$ is kept secret to the user $U$ and the registration center $R$ only.

- secrecy_of sec_b2: It represents that the server $S$'s shared secret key $B_2$ is kept secret to the user $U$ and the registration center $R$ only.

- secrecy_of sec_rks: It represents that the nonce $Rks$ generated by the server $S$ is kept secret to the server $S$ only.

- secrecy_of sec_rs: It represents that the nonce $Rs$ generated by the server $S$ is kept secret to the server $S$ only.

role alice(U,S,R :agent, C1,C2,B1,B2: symmetric_key, H,T: hash_func,
 SND, RCV: channel (dy))
played_by U
def=
local State: nat,
 Ca,VD, UID, SSID, RID, Ru, Rs, PUID, PUIDn, PSID, Rku, Rks,
 Xu, Xs, PSIDn, C1n,  D1,D1n: text,
 Ps, Pu, T1, T3, SK, E1, E2, E3, E4, E5, E6, E7, E8,
 Vur, Vsr, Vru, Vus, Vsu: text
const
 sec_vd, sec_c1, sec_c2, sec_xs, sec_xu,
 sec_rku, sec_ru, sec_uid : protocol_id,
 auth_vru, auth_vsu, auth_vus, auth_vur : protocol_id
init State := 0
transition
1. State = 0/\ RCV(start)  =|>
 State':= 1/\ secret({C1}, sec_c1, {U,R})
 /\ secret({C2}, sec_c2, {U,R}) /\ secret({Xu}, sec_xu, {R})
 /\ secret({Rku}, sec_rku, {U}) /\ secret({VD}, sec_vd, {U})
 /\ secret({Ru}, sec_ru, {U,R}) /\ secret({UID}, sec_uid, {U,R})
 /\ PUIDn':= new() /\ Rku':= new() /\ Pu':= T(Ca.Rku')
 /\ Ru':= new()  /\ E1':= xor(C1,Ru')
 /\ E2':= xor(H(Ru'.PUID.RID),UID)
 /\ E3':= xor(PUIDn',H(Ru'.UID))
 /\ Vur':= H(UID.PUID.PUIDn'.Ru'.Pu'.SSID.T1.C2)
% Send the first message to server
 /\ SND (PUID.E1'.E2'.E3'.Vur'.Pu'.T1)
 /\ witness(U, R, auth_vur, Vur')
% Receive the reply message from registration center
2. State = 1/\ RCV (E7'.E8'.H(C2.Ru'.Ps').H(B2.Rs'.Pu').Ps'.H(SK'.SSID.RID))=|>
 State':= 2/\ Vru' := H(C2.Ru'.Ps') /\ request(U, R, auth_vru, Vru')
 /\ SK' := T(Ps'.Rku)  /\ Vsu' := H(SK'.SSID.RID)
 /\ request(U, S, auth_vsu, Vsu')
 /\ C1n' := xor(E8',H(Ru'.PUID)) /\ D1n' := xor(C1n',VD)
 /\ D1' := D1n'  /\ PUID' := PUIDn /\ Vus' := H(SK'.RID)
% Send the authentication request message to server
 /\ SND(E7'.H(B2.Rs'.Pu').Vus')
 /\ witness(U, S, auth_vus, Vus')
end role

**Fig 4. Role specification in HLPSL for the user *U*.**

role bob(U,S,R :agent, C1,C2,B1,B2: symmetric_key, H,T: hash_func,
 SND, RCV: channel(dy))
played_by S
def=
local State: nat,
 Ca,UID, SSID, RID, Ru, Rs, PUID, PUIDn, PSID, Rku, Rks
 , Xu, Xs, PSIDn, B1n: text,
 Ps, Pu, T1, T3, SK, E1, E2, E3, E4, E5, E6, E7, E8,
 Vur, Vsr, Vru, Vrs, Vus, Vsu: text
const
 sec_xs, sec_b1, sec_b2, sec_rks, sec_rs, sec_sid : protocol_id,
 auth_vsr, auth_vsu, auth_vrs, auth_vus : protocol_id
init State := 0
transition
% Receive the authentication request message from alice
1. State = 0 $\wedge$ RCV(PUID.E1'.E2'.E3'.Vur'.Pu'.T1) =|>
 State':= 1 $\wedge$ secret({Xs}, sec_xs, {R}) $\wedge$ secret({B1}, sec_b1, {S, R})
 $\wedge$ secret({B2}, sec_b2, {S,R}) $\wedge$ secret({Rks}, sec_rks, {S})
 $\wedge$ secret({Rs}, sec_rs, {S,R}) $\wedge$ secret({SSID}, sec_sid, {U,S,R})
 $\wedge$ PSIDn':= new() $\wedge$ Rs':= new() $\wedge$ Rks':= new()
 $\wedge$ Ps':= T(Ca.Rks') $\wedge$ SK':= T(Pu'.Rks')
 $\wedge$ E4' := xor(B1,Rs') $\wedge$ E5' := xor(H(Rs'.PSIDn'.RID),SSID)
 $\wedge$ E6' := xor(PSIDn',H(Rs'.SSID))
 $\wedge$ Vsu':= H(SK'.SSID.RID)
 $\wedge$ Vsr' := H(SSID.PSID.PSIDn'.Rs'.Pu'.Ps'.T3.Vsu'.B2)
% Send the login request message to registration center
 $\wedge$ SND(PUID.E1'.E2'.E3'.Vur'.E4'.E5'.E6'.Vsr'.T1.T3.Ps'.Pu'.Vsu')
 $\wedge$ witness(S, R, auth_vsr, Vsr')
 $\wedge$ witness(S, U, auth_vsu, Vsu')
% Receive the login reply message from server
2. State = 1 $\wedge$ RCV(E7'.H(B2.Rs'.Pu').H(SK'.RID)) =|>
 State':= 2 $\wedge$ Vrs':=H(B2.Rs'.Pu')
 $\wedge$ request(S, R, auth_vrs, Vrs') $\wedge$ Vus' := H(SK'.RID)
 $\wedge$ request(S, U, auth_vus, Vus') $\wedge$ B1n' := xor(E7',H(Rs'.PSID))
 $\wedge$ B1' := B1n' $\wedge$ PSID' := PSIDn
end role

**Fig 5. Role specification in HLPSL for the server S.**

role server(U,S,R :agent, C1,C2,B1,B2: symmetric_key, H,T: hash_func,
           SND, RCV: channel(dy))
played_by R
def=
local      State: nat,
           VD,UID, SSID, RID, Ru, Rs, PUID, PUIDn, PSID,
           Rku, Rks, Xu, Xs, PSIDn, Rucs,Rscs: text,
           Ps, Pu, T1, T3, SK, E1, E2, E3, E4, E5, E6, E7, E8,
           Vur, Vsr, Vru, Vrs, Vus, Vsu: text
const
           sec_vd, sec_rscs, sec_rucs, sec_b1, sec_b2, sec_rks, sec_rs,
           sec_sid, sec_c1, sec_c2, sec_xs, sec_xu, sec_rku, sec_ru, sec_uid: protocol_id,
           auth_vsr, auth_vsu, auth_vrs, auth_vus, auth_vru, auth_vur : protocol_id
init   State := 0
transition
1. State = 0 $\wedge$ RCV(PUID.E1'.E2'.E3'.H(UID.PUID.PUIDn'.Ru'.Pu'.SSID.T1.C2).
        E4'.E5'.E6'.H(SSID.PSID.PSIDn'.Rs'.Pu'. Ps'.T3.Vsu'.B2).
        T1.T3.Ps'.Pu'.H(SK'.SSID.RID)) =|>
  State':= 1$\wedge$ secret({C1}, sec_c1, {U,R})  $\wedge$ secret({C2}, sec_c2, {U,R})
        $\wedge$  secret({VD}, sec_vd, {U}) $\wedge$ secret({Xu}, sec_xu, {R})
        $\wedge$ secret({Rku}, sec_rku, {U}) $\wedge$ secret({Ru}, sec_ru, {U,R})
        $\wedge$ secret({UID}, sec_uid, {U,R}) $\wedge$ secret({Xs}, sec_xs, {R})
        $\wedge$ secret({B1}, sec_b1, {S, R}) $\wedge$ secret({B2}, sec_b2, {S,R})
        $\wedge$ secret({Rks}, sec_rks, {S}) $\wedge$ secret({Rs}, sec_rs, {S,R})
        $\wedge$ secret({SSID}, sec_sid, {U,S,R}) $\wedge$ secret({Rscs}, sec_rscs, {R})
        $\wedge$ secret({Rucs}, sec_rucs, {R})
        $\wedge$ Ru':= xor(E1',H(PUID.RID.Xu)) $\wedge$ UID' := xor(E2',H(Ru'.PUID.RID))
        $\wedge$ C2' := H(UID'.Xu.Rucs) $\wedge$ PUIDn' := xor(E3',H(Ru'.UID'))
        $\wedge$ Rs':= xor(E4',H(PSID.RID.Xs)) $\wedge$ SSID' := xor(E5',H(Rs'.PSID.RID))
        $\wedge$ B2' := H(SSID'.Xs.Rscs) $\wedge$ PSIDn' := xor(E6',H(Rs'.SSID'))
        $\wedge$ Vur' := H(UID.PUID.PUIDn'.Ru'.Pu'.SSID.T1.C2)
        $\wedge$ request(R, U, auth_vur, Vur')
        $\wedge$ Vsr' := H(SSID.PSID.PSIDn'.Rs'.Pu'.Ps'.T3.Vsu'.B2)
        $\wedge$ request(R, S, auth_vsr, Vsr')
        $\wedge$ E7':= xor(H(PSIDn'.RID.Xs),H(Rs'.PSID)) $\wedge$ Vrs' := H(B2'.Rs'.Pu')
        $\wedge$ E8':= xor(H(PUIDn'.RID.Xu),H(Ru'.PUID)) $\wedge$ Vru' := H(C2.Ru'.Ps')
        $\wedge$ SND(E7'.E8'.Vru'.Vrs'.Ps'.H(SK'.SSID.RID))
        $\wedge$ witness(R, U, auth_vru, Vru')
        $\wedge$ witness(R, S, auth_vrs, Vrs')
end role

**Fig 6. Role specification in HLPSL for the registration center *R*.**

% ======================== session ========================
role session(U,S,R :agent,C1,C2,B1,B2: symmetric_key, H,T: hash_func)
def=
local R1, R2, R3: channel(dy)
composition
        alice(U, S, R,  C1,C2, B1, B2, H,T, R1,R3)
        /\ bob(U, S, R,  C1,C2, B1, B2, H,T, R2,R1)
        /\ server(U, S, R,  C1,C2, B1, B2, H,T, R3,R2)
end role

**Fig 7. Role specification in HLPSL for the session.**

- secrecy_of sec_sid: It represents that the server $S$'s identity $SID$ is kept secret to the user $U$, the server $S$ and the registration center $R$ only.

- secrecy_of sec_c1: It represents that the user $U$'s shared secret key $C_1$ is kept secret to the user $U$ and the registration center $R$ only.

- secrecy_of sec_c2: It represents that the user $U$'s shared secret key $C_2$ is kept secret to the user $U$ and the registration center $R$ only.

- secrecy_of sec_xu: It represents that registration center $R$'s secret key $X_U$ is kept secret to $R$ only.

- secrecy_of sec_rku: It represents that the nonce $Rku$ generated by the user $U$ is kept secret to the user $U$ only.

% ======================== environment ========================
role environment()
def=
const u, s, r: agent, c1,c2,b1,b2,a1,a2: symmetric_key,h: hash_func,t: hash_func,
        sec_vd, sec_rscs, sec_rucs, sec_b1, sec_b2, sec_rks, sec_rs, sec_sid
        , sec_c1, sec_c2, sec_xs, sec_xu, sec_rku, sec_ru, sec_uid: protocol_id,
        auth_vsr, auth_vsu, auth_vrs, auth_vus, auth_vru,  auth_vur : protocol_id
intruder_knowledge = {u, s, r, a1, a2, h, t}
composition
        session(u, s, r, c1, c2, b1, b2,h,t)
        /\ session(u, i, r, c1,c2,a1,a2, h,t)
        /\ session(i, s, r, a1,a2,b1,b2, h,t)
        /\ session(u, s, i, c1, c2, b1, b2, h,t)
end role

**Fig 8. Role specification in HLPSL for the environment.**

```
% ========================= goal ============
goal
secrecy_of sec_rscs
secrecy_of sec_vd
secrecy_of sec_rucs
secrecy_of sec_xs
secrecy_of sec_b1
secrecy_of sec_b2
secrecy_of sec_rks
secrecy_of sec_rs
secrecy_of sec_sid
secrecy_of sec_c1
secrecy_of sec_c2
secrecy_of sec_xu
secrecy_of sec_rku
secrecy_of sec_ru
secrecy_of sec_uid
authentication_on auth_vsr
authentication_on auth_vsu
authentication_on auth_vrs
authentication_on auth_vus
authentication_on auth_vru
authentication_on auth_vur
end goal
environment()
```

**Fig 9. Role specification in HLPSL for the goal.**

- secrecy_of sec_ru: It represents that the nonce *Ru* generated by the user *U* is kept secret to the user *U* only.

- secrecy_of sec_uid: It represents that the user *U*'s identity *UID* is kept secret to the user *U* and the registration center *R* only.

- authentication_on auth_vsr: It represents that the registration *R* authenticates the server *S*.

- authentication_on auth_vsu: It represents that the user *U* authenticates the server *S*.

- authentication_on auth_vrs: It represents that the server *S* authenticates the registration center *R*.

- authentication_on auth_vus: It represents that the server *S* authenticates the user *U*.

- authentication_on auth_vru: It represents that the user *U* authenticates the registration center *R*.

% ============== result OFMC  ============

% OFMC
% Version of 2006/02/13
SUMMARY
  SAFE
DETAILS
  BOUNDED_NUMBER_OF_SESSIONS
PROTOCOL
  /home/span/span/testsuite/results/myprotocol.if
GOAL
  as_specified
BACKEND
  OFMC
COMMENTS
STATISTICS
  parseTime: 0.00s
  searchTime: 0.64s
  visitedNodes: 55 nodes
  depth: 6 plies

**Fig 10. The result of the analysis using OFMC back-end.**

- authentication_on auth_vur: It represents that the registration center $R$ authenticates the user $U$.

**Analysis of the results.**   We have simulated the proposed scheme using FMC and CL-AtSe back-ends of AVISPA. The simulation results for the security verification are shown in Figs 10 and 11.

The results ensure that the proposed scheme is secure under the test of AVISPA using OFMC and CL-AtSe back-ends, and guarantees user anonymity, and it is also secure against the passive attacks and the active attacks, such as the replay attack and man-in-the-middle attack.

### 5.3 Informal security analysis

In this part, we demonstrate the proposed scheme can resist various kinds of attacks.

**Mutual authentication.**   The proposed scheme provides the mutual authentication.

In the step 3 of the authentication phase, the registration center $R$ computes $r_U = E_1 \oplus h(PUID_i||RID||x_U)$, $UID_i = E_2 \oplus h(r_U||PUID_i||RID)$, $UD_2^{*'} = h(UID_i||x_U||RU_{cs})$, $PUID_i^{new} = E_3 \oplus h(r_U||UID_i)$, $r_S = E_4 \oplus h(PSID_j||RID||x_S)$, $SID_j = E_5 \oplus h(r_S||PSID_j||RID)$, $SD_2^* = h(SID_j||x_S||RS_{cs})$,

% ================ result CL-AtSe ================

SUMMARY
  SAFE

DETAILS
  BOUNDED_NUMBER_OF_SESSIONS
  TYPED_MODEL

PROTOCOL
  /home/span/span/testsuite/results/myprotocol.if

GOAL
  As Specified

BACKEND
  CL-AtSe

STATISTICS

  Analysed   : 1303 states
  Reachable  : 325 states
  Translation: 8.36 seconds
  Computation: 0.11 seconds

**Fig 11. The result of the analysis using CL-AtSe back-end.**

$PSID_j{}^{new} = E_6 \oplus h(r_S || SID_j)$ and checks if the user's identity $UID_i$ is included in the verification table. If so, $R$ computes $V_{UR}' = h(UID_i || PUID_i || PUID_i{}^{new} || r_U || P_U || SID_j || T_1 || UD_2{}^*)$ and checks if $V_{UR}' = V_{UR}$. If so, $R$ authenticates the user $UR_i$. The proof of this authentication is as follows. At first, $UD_2{}^*$ included in $V_{UR}$ is a secret key known to only the user $UR_i$ and the registration center $R$. And $V_{UR}$ contains the nonce $r_U$ generated by the user. Thus the registration center can verify that $V_{UR}$ was sent by the user and that it wasn't replayed if $V_{UR}' = V_{UR}$ is true.

And the registration center $R$ checks if the server's identity $SID_j$ is included in the verification table. If so, $R$ computes $V_{SR}' = h(SID_j || PSID_j || PSID_j{}^{new} || r_S || P_U || P_S || T_3 || V_{SU} || SD_2)$ and checks if $V_{SR}' = V_{SR}$. If so, $R$ authenticates the server $SR_j$. The proof of this authentication is as follows. At first, $SD_2$ included in $V_{SR}$ is a secret key known to only the server $SR_j$ and the registration center $R$. And $V_{SR}$ contains the nonce $r_S$ generated by the server. Thus the registration center can verify that $V_{SR}$ was sent by the server and that it wasn't replayed if $V_{SR}' = V_{SR}$ is true.

In the step4, the user $UR_i$ computes $V_{RU}' = h\,(UD_2^*||r_U||P_S)$ and checks if $V_{RU}' = V_{RU}$. If so, $UR_i$ authenticates the registration center $R$. The proof of this authentication is as follows. At first, $UD_2^*$ included in $V_{RU}$ is a secret key known to only the user $UR_i$ and the registration center $R$. And $V_{RU}$ contains the nonce $r_U$ generated by the user. Thus the user can verify that $V_{RU}$ was sent by $R$ and that it wasn't replayed if $V_{RU}' = V_{RU}$ is true.

And $UR_i$ computes $SK = T_{rkU}(P_S) = T_{rkSrkU}(\alpha) \bmod p$, $V_{SU}' = h(SK||SID_j||RID)$ and authenticates the server $SR_j$ by checking if $V_{SU}' = V_{SU}$. The proof of this authentication is as follows. Because $SK$ included in $V_{SU}$ is computed as $SK = T_{rkS}(P_U) = T_{rkU}(P_S) = T_{rkSrkU}(\alpha) \bmod p$, it is a secret generated by only the server $SR_j$ except for the user $UR_i$. Also it contains the nonce $r_{kU}$ generated by the user in step 1. Thus the user can verify that $V_{SU}$ was sent by the server and that it wasn't replayed if $V_{SU}' = V_{SU}$ is true.

In the step5, the server computes $V_{RS}' = h\,(SD_2||r_S||P_U)$ and checks if $V_{RS}' = V_{RS}$. If so, the server authenticates the registration center $R$. The proof of this authentication is as follows. At first, $SD_2$ included in $V_{RS}$ is a secret key known to only the server $SR_j$ and the registration center $R$. And $V_{RS}$ contains the nonce $r_S$ generated by the server. Thus the user can verify that $V_{RS}$ was sent by $R$ and that it wasn't replayed if $V_{RS}' = V_{RS}$ is true.

And the server computes $V_{US}' = h(SK||RID)$ and authenticates the user $UR_i$ by checking if $V_{US}' = V_{US}$. The proof of this authentication is as follows. Because $SK$ included in $V_{US}$ is computed as $SK = T_{rkS}(P_U) = T_{rkU}(P_S) = T_{rkSrkU}(\alpha) \bmod p$, it is a secret generated by only the user $UR_i$ except for the server $SR_j$. Also it contains the nonce $r_{kS}$ generated by the server in step2. Thus the server can verify that $V_{US}$ was sent by the user and that it wasn't replayed if $V_{US}' = V_{US}$ is true.

Therefore, the registration center authenticates the user and server in the step3, the user authenticates the registration center and server in step4, and the server authenticates the registration center and user in step5. Thus the proposed scheme achieves the mutual authentication between the registration center, user and server.

**User anonymity.** The proposed scheme provides user anonymity for key exchange.

The data that an attacker can use to get the user's identity $UID_i$ is $E_2 = h(r_U||PUID_i||RID) \oplus UID_i$ among the messages $M_1 = \{PUID_i, E_1, E_2, E_3, V_{UR}, P_U, T_1\}$, $M_2 = \{PUID_i, E_1, E_2, E_3, V_{UR}, E_4, E_5, E_6, V_{SR}, T_1, T_3, P_S, V_{SU}\}$, $M_3 = \{E_7, E_8, V_{RU}, V_{RS}, P_S, V_{SU}\}$ and $M_4 = \{E_7, V_{RS}, V_{US}\}$ in authentication key exchange process. If the attacker wants to get $UID_i$, he may compute as follows: $UID_i = E_2 \oplus h\,(r_U||PUID_i||RID)$. For this, the attacker needs to know the nonce $r_U$ generated by the user and has to compute $r_U = E_1 \oplus h\,(PUID_i||RID||x_U)$. So the attacker also needs to know $x_U$, but the attacker cannot get $x_U$ because it is a secret known to only the registration center $R$. Thus, the attacker cannot get the user's identity $UID_i$.

If the attacker wants to get the server's identity $SID_j$, he may compute as follows: $SID_j = E_5 \oplus h(r_S||PSID_j||RID)$. For this, the attacker needs to know the nonce $r_S$ generated by the server and has to compute $r_S = E_4 \oplus h(PSID_j||RID||x_S)$. So the attacker also needs to know $x_S$ but the attacker cannot get $x_S$ because it is a secret known to only the registration center $R$. Thus, the attacker cannot get the server's identity $SID_j$.

As a result, the attacker cannot get both the user's identity and server's identity.

**Perfect forward security of session key.** In the proposed scheme, the session key $SK$ is computedd as $SK = T_{rkS}(P_U) = T_{rkU}(P_S) = T_{rkSrkU}(\alpha) \bmod p$. It contains the random numbers $rk_S$ and $rk_U$ generated by the different session entities for each session. Thus, even if the attacker gets $rk_S$ and $rk_U$ for the current session, he cannot compute the session key for the previous session. Therefore, the proposed scheme provides the perfect forward secrecy of session key.

**Untraceability.** The proposed scheme provides the untraceability.

Let's imagine that the attacker can get the secret data of the user and server for the previous session by stealing previous messages. But in the step 4 of authentication key exchange phase,

the user replaces ($UD_1$, $PUID_i$) with ($UD_1^{new}$, $PUID_i^{new}$) and in the step 5, the server replaces ($SD_1$, $PSID_j$) with ($SD_1^{new}$, $PSID_j^{new}$). And for the next session, both the user and server combine their identities with the updated secret data and generated new random numbers. Therefore, the attacker cannot get the identities of the user and server, for the next session using the previous messages, so he cannot know the current communicating entities.

**No key control.**   The proposed scheme provides no key control property.

In the proposed scheme, the session key $SK$ is computed as $SK = T_{rkS}(P_U) = T_{rkU}(P_S) = T_{rkSrkU}(\alpha) \bmod p$. In this equation, $P_S$ is computed as $T_{rkS}(\alpha) \bmod p$ and $P_U$ is computed as $T_{rkU}(\alpha) \bmod p$. And $rk_S$ is only known to the server and the user knows only $P_S$. But according to the rules CDLP [36] and CDHP [36], the user never computes $rk_S$ from $P_S$. Also $rk_U$ is only known to the user and the server never gets $rk_U$ from $P_U$. Thus, the session key cannot be generated by each of the user and server, and it can be only generated by the agreement of both of them.

**Off-line password guessing attack.**   The proposed scheme resists the password guessing attack.

This scheme does not use passwords during the authentication process but only uses passwords for access to the smart card. The information stored in the user's smart card is ($UD_1$, $UD_2$, $UD_3$, $PUID_i$, $RID$, $P_i$) and the information that can be used for guessing password is $UD_1 = UD_1^*$ $\oplus h(PW_i\|R_i\|UID_i)$ and $UD_2 = UD_2^* \oplus h(PW_i\|R_i\|UID_i)$. Let's imagine that an attacker steals the user's smart card $SC_i$ and gets his identity $UID_i$. Then to guess the password $PW_i$, the attacker must compute $VD_i^* = h(PW_i^*\|R_i\|UID_i)$, $UD_1^{*'} = UD_1 \oplus VD_i^*$, $UD_2^{*'} = UD_2 \oplus VD_i^*$, $UD_3' = h(VD_i^*\|UD_1^{*'}\|UD_2^{*'})$ by using $UID_i$ and any password $PW_i^*$ to compare $UD_3'$ and $UD_3$ stored in $SC_i$. But the attacker cannot get $R_i$ because he cannot know the user's biometric $BIO_i$ so he cannot calculate above equations. Thus, the attacker cannot guess the user's password.

**Privileged insider attack.**   The proposed scheme is secure against the privileged-insider attack. In the registration phase of the proposed scheme, only the user's identifier is transmitted to the registration center through a secure channel and the user's password and biometric are not transmitted to the registration center. Therefore, the privilege insider of the registration center cannot know the user's password and biometric. Therefore, the proposed scheme is secure against this attack.

**Stolen verifier attack.**   The proposed scheme is secure against stolen verifier attack.

In the registration phase, the registration center $R$ stores {$UID_i$, $RU_{cs}$} in the user registration table. Here $UID_i$ is the identity of the user $UR_i$ and $RU_{cs}$ is the random number chosen by $R$. The essential factors that $R$ can use to authenticate the user are the shared secrets between the registration center and user, $UD_1^* = h(PUID_i\|RID\|x_U)$, $UD_2^* = h(UID_i\|x_U\|RU_{cs})$ and the random numbers $r_U$, $rk_U$ generated by the user. So even if the attacker knows $UID_i$ and $RU_{cs}$, he cannot pass the authentication steps safely. Therefore, the attacker cannot be successful in this attack.

**User impersonate attack.**   The proposed scheme is secure against the user impersonate attack. In order to impersonate as the user $UR_i$, the attacker has to compute $E_1 = UD_1^* \oplus r_U$, $E_2 = h(r_U\|PUID_i\|RID) \oplus UID_i$ and $E_3 = PUID_i^{new} \oplus h(r_U\|UID_i)$. Let's imagine that the attacker knows $PUID_i$, $RID$, $UID_i$ and he generates $PUID_i^{new}$ and a nonce $r_U$. Then he can calculate $E_3 = PUID_i^{new} \oplus h(r_U\|UID_i)$ and $E_2 = h(r_U\|PUID_i\|RID) \oplus UID_i$. But he cannot compute $E_1 = UD_1^* \oplus r_U$ without knowing of $UD_1^*$. But he cannot compute $UD_1^* = h(PUID_i\|RID\|x_U)$ because $x_U$ is a secret known to only the registration center and cannot also calculate $E_1 = UD_1^* \oplus r_U$.

Therefore, the attacker cannot impersonate as $UR_i$ and achieve this attack.

**Server impersonate attack.**   The proposed scheme is secure against the server impersonate attack.

In order to impersonate as the server $SR_j$, the attacker has to compute $E_4 = SD_1 \oplus r_S$, $E_5 = h(r_S||PSID_j^{new}||RID) \oplus SID_j$ and $E_6 = PSID_j^{new} \oplus h(r_S||SID_j)$. Let's imagine that the attacker knows $RID$, $SID_j$ and he generates $PSID_j^{new}$ and a nonce $r_S$. Then he can calculate $E_6 = PSID_j^{new} \oplus h(r_S||SID_j)$ and $E_5 = h(r_S||PSID_j^{new}||RID) \oplus SID_j$. But he cannot compute $E_4 = SD_1 \oplus r_S$ without knowing of $SD_1$. But he cannot compute $SD_1 = h(PSID_j||RID||x_S)$ because $x_S$ is a secret known to only the registration center and cannot also calculate $E_4 = SD_1 \oplus r_S$.

Therefore, the attacker cannot impersonate as $SR_j$ and achieve this attack.

**Man-in-the-middle attack.** As it is shown above, the proposed scheme achieves certain mutual authentication and the attacker can neither impersonate as the initiator $UR_i$ and the responder $SR_j$, so an attacker cannot achieve the man-in-the-middle attack. The reasons for this are as follows.

First, the attacker cannot exchange any messages with the user by impersonating as the responder, valid server. As we show above, the attacker cannot compute $SD_1 = h(PSID_j||RID||x_S)$ because $x_S$ is a secret known to only the registration center and cannot also calculate $E_4 = SD_1 \oplus r_S$. Hence, the attacker cannot impersonate as the responder. Also the attacker cannot exchange any messages with the server by impersonating as the initiator, valid user. As we show above, the attacker cannot compute $UD_1^* = h(PUID_i||RID||x_U)$ because $x_U$ is a secret known to only the registration center and cannot also calculate $E_1 = UD_1^* \oplus r_U$. Hence, the attacker cannot impersonate as the initiator.

In conclusion, the attacker cannot achieve the man-in-the-middle attack.

**Replay attack.** In the step 2 of the authentication key exchange phase, after receiving the message $M_1$, the server checks if $\Delta T = T_1 - T_2 < \Delta T_{define}$ and if it is false, the server stops the session. Here $T_1$ is the time when the message is sent and $T_2$ is the time when the message is received. So the replay attack can't be achieved in this step. Also in the step 3, the attacker cannot achieve the replay attacker because the message $M_3$ contains a time stamp. In the step 4, the user checks if $V_{RU} = h(UD_2^*||r_U||P_S)$ is true. Here $r_U$ is a nonce generated by the user in the step 1, so in the case that the attacker replays the message $M_3$ to the user, the user can recognize this attack using $r_U$. Like this, in the step 5, the server can recognize that the message $M_4$ was replayed by the attacker by checking $r_S$ generated by the server in the step 2.

Therefore, the attacker cannot achieve replay attack.

**Forgery attack.** Forgery attack means that an attacker attempts to forge captured messages to masquerade as the legitimate user for wireless system access to the resources.

These followings are the analysis of messages in the proposed scheme.

- In the message $M_1$, $E_1$ and $V_{UR}$ both contain $x_U$.

- In $M_2$, $x_S$ is required in both $E_4$ and $V_{SR}$, besides the original elements in $M_1$.

- In $M_3$, $E_7$, $V_{RS}$, $E_8$ and $V_{RU}$ all needs $x_U$ and $x_S$.

- In $M_4$, $x_U$ and $x_S$ are required in both $E_7$ and $V_{RS}$.

As it is shown above, the attacker has to know both $x_U$ and $x_S$ to forge any messages in the session. But $x_U$ and $x_S$ cannot be captured by the attacker because both of them are the secret keys known to only the registration center $R$. Therefore, the attacker cannot forge any messages and we can claim that the proposed scheme resists forgery attack.

**Known key security.** In the proposed scheme, the session key $SK$ is calculated as $SK = T_{rkS}(P_U) = T_{rkU}(P_S) = T_{rkSrkU}(\alpha) \bmod p$. It contains the random numbers $rk_S$ and $rk_U$ that are generated by session entities for each session. Even if an attacker gets the previous session key, he cannot calculate the current session key.

Therefore, the proposed scheme provides known key security property.

**Table 4. Comparison of the computational cost between the proposed scheme and other schemes in the authentication and session key exchange phase.**

|  | Lwamo et al. [12] | Zhou et al. [31] | Tomar et al. [35] | proposed |
|---|---|---|---|---|
| $UR_i$ | $9t_h+t_s+t_p$ | $10t_h$ | $3t_e + 11t_h$ | $2t_c + 8t_h$ |
| $SR_j$ | $9t_h+ 2t_s+ t_p$ | $7t_h$ | $3t_e + 7t_h$ | $2t_c + 7t_h$ |
| $R$ |  | $19t_h$ | $2t_e + 12t_h$ | $16t_h$ |
| Total | $18t_h+3t_s+ 2t_p$ | $36t_h$ | $8t_e+ 30t_h$ | $4t_c + 31t_h$ |
| Total execution time | 7.7552ms | 0.0828ms | 17.877ms | 8.9753ms |
| Round | 3 | 4 | 5 | 4 |

## 6. Performance comparisons

In this section, we compare the computational cost, communication overhead and security performance of the proposed scheme with the recent similar authentication key exchange protocols [12, 31, 35].

The notations used for comparison of computational cost are as follows.

$t_c$: time needed for Chebyshev polynomial operation

$t_e$: time needed for a scalar multiplication on elliptic curve

$t_s$: time needed for symmetric encryption/decryption operation

$t_p$: time needed for public key encryption/decryption operation

$t_h$: time needed for one-way hash function operation

Table 4 shows the comparison of the computational cost of the four schemes, including the proposed scheme in the authentication and session key exchange phase. According to the execution overhead given in [50–52], in the environment where CPU is 2.20GHz and RAM is 2048MB, it takes about 0.0023ms, 0.0046ms, 2.226ms, 3.85ms, 2.226ms to execute the one-way hash function, symmetric encryption/decryption, the scalar multiplication on elliptic curve, public key encryption/decryption and Chebyshev polynomial operation respectively. Compared with other schemes, the result shows that our scheme requires nearly low computational cost.

In order to measure the communication overhead of our proposed scheme, let us assume the bit size of identity, random number, timestamp, hash output, Chebyshev chaotic maps and elliptic curve cryptography as $|ID| = 160$, $|N| = 160$, $|Ts| = 32$, $|H| = 160$, $|T| = 160$ and $|E| = 320$ bits respectively.

Table 5 shows the communication overhead of our proposed scheme according to above assumption.

Table 6 shows the comparison of the communication overhead of our proposed scheme and three other schemes. As shown in Table 6, the communication overhead of our proposed scheme is higher than other schemes.

**Table 5. Communication overhead of our proposed scheme.**

|  | Expression | Length of message(bits) |
|---|---|---|
| M1 | $|ID| +4|H| + |T|+|Ts|$ | 992 |
| M2 | $|ID|+9|H| +2|Ts|+2|T|$ | 1984 |
| M3 | $5|H| +|T|$ | 960 |
| M4 | $3|H|$ | 480 |
| Total | $2|ID| + 21|H| + 4|T|+3|Ts|$ | 4416 |

**Table 6. Comparison of the computational cost between the proposed scheme and other schemes.**

|  | Lwamo et al. [12] | Zhou et al. [28] | Tomar et al. [32] | proposed |
|---|---|---|---|---|
| M1 | 2\|H\|+2\|ID\|+\|Ts\| | \|ID\|+4\|H\| | \|ID\|+4\|H\|+\|E\|+\|Ts\| | \|ID\| +4\|H\| + \|T\|+\|Ts\| |
| M2 | 3\|N\|+3\|ID\|+\|H\|+2\|Ts\| | 8\|H\|+2\|ID\| | \|ID\|+7\|H\|+2\|E\|+\|Ts\| | \|ID\|+9\|H\| +2\|Ts\|+2\|T\| |
| M3 | \|H\| | 6\|H\| | 3\|H\|+\|Ts\| | 5\|H\| +\|T\| |
| M4 |  | 3\|H\| | 3\|H\|+\|E\|+\|Ts\| | 3\|H\| |
| M5 |  |  | \|H\| |  |
| Total | 5\|ID\|+3\|N\|+4\|H\|+3\|Ts\| | 3\|ID\|+21\|H\| | 2\|ID\|+18\|H\|+4\|E\|+4\|Ts\| | 2\|ID\| + 21\|H\| + 4\|T\|+3\|Ts\| |
| Total bits | 2016 | 3840 | 4608 | 4416 |

Table 7 shows the comparative evaluation of the security function between the proposed scheme and other schemes.

As shown in Tables 4, 6 and 7, the proposed scheme outperforms the other schemes in terms of the security properties presented.

Lwamo et al.'s scheme has high computational cost because it uses the public key encryption. Also his scheme has lower communication overhead than ours' scheme and doesn't provide re-registration with the original identity and it is vulnerable to the stolen smart card attack.

Zhou et al.'s scheme uses only hash functions so it has very low computational cost, but it has lower communication overhead than ours. And it is vulnerable to the replay attack and doesn't provide various properties such as mutual authentication, no key control, re-registration with the original identity, and efficiency in the verification of wrong password.

Tomar et al.'s scheme provides the security properties mentioned in the Table 7 but his scheme has higher computational cost and higher communication overhead than our proposed scheme. And it doesn't provide the re-registration with the original identity.

As shown in Tables 4, 6 and 7, the schemes with strong security performances have high computational cost, while the schemes with low computational cost don't provide the strong security performances.

**Table 7. Comparative evaluation of the security function between the proposed scheme and other schemes.**

|  | Lwamo et al. [12] | Zhou et al. [31] | Tomar et al. [35] | proposed |
|---|---|---|---|---|
| Provision of mutual authentication | Yes | No | Yes | Yes |
| Provision of User anonymity | Yes | Yes | Yes | Yes |
| Provision of untraceability | Yes | Yes | Yes | Yes |
| Protection of password guessing attack | Yes | Yes | Yes | Yes |
| Protection of Privileged insider attack | Yes | Yes | Yes | Yes |
| Protection of stolen smart card attack | No | Yes | Yes | Yes |
| Protection of User impersonate attack | Yes | Yes | Yes | Yes |
| Efficiency in the verification of wrong password | Yes | No | Yes | Yes |
| Protection of replay attack | Yes | No | Yes | Yes |
| Provision of no key control | Yes | No | Yes | Yes |
| Reregistration with the original identity | No | No | No | Yes |
| Using biometric | Yes | Yes | Yes | Yes |
| Using smart card | Yes | Yes | Yes | Yes |

## 7. Conclusion

In this work, we analysed Lwmao et al.'s scheme and Zhou et al.'s scheme, pointed out its weakness and proposed an improved chaotic mapping-based authentication key agreement protocol with low computational cost, high communication overhead, robust security performance and strong mutual authenticaton. The proposed scheme was designed to provide strong mutual authentication between communication participants, so the length of messages is long and communication overhead is relatively high. In the proposed scheme, we allowed the users to re-register without modifying their identities by including the random numbers in their secret keys shared with the registration center in the registration phase. Also, we used the users' biometrics and the fuzzy extractor to keep their privacies more secure. We also prevented the replay attack using timestamps and chaotic maps, and provided the robust mutual authentication and safer session key agreement. The proposed scheme also achieved various security properties and attack resistances such as the anonymity, untraceability and resistance of stolen smart card attack. Also, we formally analysed our protocol based on BAN logic and AVISPA tool, and demonstrated that it is secure against various attacks through informal security analysis.

## Author Contributions

**Conceptualization:** Hyang-Rim Jo, Kyong-Sok Pak.

**Data curation:** Il-Jin Zhang.

**Formal analysis:** Hyang-Rim Jo, Chung-Hyok Kim.

**Investigation:** Chung-Hyok Kim.

**Project administration:** Kyong-Sok Pak.

**Software:** Hyang-Rim Jo, Chung-Hyok Kim, Il-Jin Zhang.

**Supervision:** Kyong-Sok Pak.

**Validation:** Hyang-Rim Jo, Il-Jin Zhang.

**Writing – original draft:** Hyang-Rim Jo, Kyong-Sok Pak, Chung-Hyok Kim, Il-Jin Zhang.

**Writing – review & editing:** Kyong-Sok Pak.

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
