## [Decision Letter · Decision Letter 0]

9 May 2022

PONE-D-22-11432Cryptanalysis and improved mutual authentication key agreement protocol using pseudo-identityPLOS ONE

Dear Dr. Pak,

Thank you for submitting your manuscript to PLOS ONE. After careful consideration, we feel that it has merit but does not fully meet PLOS ONE’s publication criteria as it currently stands. Therefore, we invite you to submit a revised version of the manuscript that addresses the points raised during the review process.

We look forward to receiving your revised manuscript.

Kind regards,

Pandi Vijayakumar, Ph.D

Academic Editor

PLOS ONE

Journal Requirements:

"No"

4. We note you have included a table to which you do not refer in the text of your manuscript. Please ensure that you refer to Tables 1 and 2 in your text; if accepted, production will need this reference to link the reader to the Table.

Additional Editor Comments:

The paper has many serious problems. So the authors should carefully address the reviewers comments and they should submit a revised version in the next round.

Reviewers' comments:

Reviewer's Responses to Questions

**Comments to the Author**

1. Is the manuscript technically sound, and do the data support the conclusions?

Reviewer #1: Yes

Reviewer #2: No

2. Has the statistical analysis been performed appropriately and rigorously? 

Reviewer #1: N/A

Reviewer #2: No

3. Have the authors made all data underlying the findings in their manuscript fully available?

Reviewer #1: Yes

Reviewer #2: No

4. Is the manuscript presented in an intelligible fashion and written in standard English?

Reviewer #1: Yes

Reviewer #2: Yes

5. Review Comments to the Author

Reviewer #1: The authors have developed a mutual key agreement scheme to be used in various applications.

They have also analysed the performance of the proposed scheme using BAN logic and AVISPA tools. The authors must address the following comments before submitting the next version of the manuscript.

The abstract must contain the name of the technical algorithm which the authors had used when comparing their work with the other two different schemes given in the abstract.

The introduction section must be strengthened. It must cover the scope of the proposed work and the objectives must be clearly defined.

Moreover, the proposed mutual key agreement scheme given in the manuscript must cover the following manuscripts. The following manuscript can be considered for analysis in the related work or for comparison in the performance analysis section. The titles of some notable referecnes are given below

Secure multifactor authenticated key agreement scheme for industrial IoT.

An efficient group key agreement protocol for secure P2P communication.

An efficient anonymous authentication and key agreement scheme with privacy-preserving for smart cities.

A provably secure dynamic ID-based authenticated key agreement framework for mobile edge computing without a trusted party.

An unlinkable authenticated key agreement with collusion resistant for VANET’s

A PUF-based lightweight authentication and key agreement protocol for smart UAV networks.

Based on the above considerations, I would recommend MAJOR revision.

Reviewer #2: In this proposed work, the authors mentioned about the Stolen smart card attack. However, practically, the smart card data should be available in the encrypted form with the support of the private key of the user like the ATM cards. In ATM cards, the the pin number is linked with the card number. Similarly, in such type of smart cards also, the data should be protected with the support of the users private keys.

In User revocation phase, the communication is completely happened in the secure channel like SSL. It will enhance the computational overhead enormously.

The authors mentioned that the proposed scheme is secure against the user impersonate attack and the forgery attack. However, there is no proper explanation available about the withstanding against the forgery. There is no proper justification against the main in the middle attack.

In performance analysis, the simulation tools are not mentioned. How did they get the timing values for the cryptographic operations without mentioning the simulation tools.

6. PLOS authors have the option to publish the peer review history of their article (what does this mean?). If published, this will include your full peer review and any attached files.

Reviewer #1: No

Reviewer #2: No

---

## [Author Response · Author response to Decision Letter 0]

22 Jun 2022

Responses to the reviewer1

1) You pointed that “The abstract must contain the name of the technical algorithm which the authors had used when comparing their work with the other two different schemes given in the abstract.” in your comments.

In our paper, we have used the experiment results of Odelu et al.’s scheme and Kilinc et al.’s scheme when comparing the computational cost of our proposed scheme with other related schemes. And we describe about this in the abstract and Section 6 of our manuscript.

2) You also pointed that “The introduction section must be strengthened.”

We added a more detailed description of the scope and purpose of the study in the introduction section. We also studied the papers you suggested and cited them and included them in the related work.

Responses to the reviewer2

1) You pointed that the communication was completely happened through the secure channel in user revocation phase.

If a user loses or is stolen by a malicious user, the user must revocate the already registered user at the user revocation phase.

Since the user information in the proposed protocol is stored on a smart card, we argue that the user should work on a secure channel in the revocation phase as well as in the registration phase.

2) You pointed that there was no proper explanation available about the withstanding against the forgery attack.

We considered the same type of attack as the impersonate attack and forgery attack in the previous manuscript. In this revised manuscript, we individually analysed the forgery attack resistance by distinguishing it from impersonate attack resistance. In the informal security analysis section, we analysed the resistance of the proposed scheme to forgery attacks. We describe about this in lines from 930 to 941.

3) You pointed that there was no proper justification against the man-in-the-middle attack.

We have analysed in detail the man-in-the-middle attack resistance of the proposed scheme in the revised manuscript. We focused on the fact that the man-in-the-middle attack was largely composed of the initiator impersonate attack and the responder impersonate attack, and revealed the attack resistance against it.In our protocol, the session initiator is just the user and the session responder is the server. We have analysed the resistance of our proposed scheme against the man-in-the-middle-attack in the informal security analysis section. We describe about this in lines from 904 to 918.

4) You pointed that the simulation tools were not mentioned in performance analysis.

In the previous manuscript, we used the labels we defined without numerically comparing the computational cost. In the revised manuscript, we use the experimental results of the previous researchers to calculate the computational cost more accurately, based on which we compare the computational cost with the previous schemes. In our paper, we have used the experiment results of Odelu et al.’s scheme, Kilinc et al.’s scheme and Jangirala et al’s scheme to compare the computational cost of our proposed scheme with other related schemes. We describe about it in lines from 945 to 967.

Additionally, we also compare the communication overhead with previous works and describe it in line from 968 to 980.

We hope that the revised manuscript meets the requirements of the editors and reviewers.

Thank you and best regards.

from Kyongsok Pak

---

## [Decision Letter · Decision Letter 1]

8 Jul 2022

Cryptanalysis and improved mutual authentication key agreement protocol using pseudo-identity

PONE-D-22-11432R1

Dear Dr. Pak,

We’re pleased to inform you that your manuscript has been judged scientifically suitable for publication and will be formally accepted for publication once it meets all outstanding technical requirements.

Kind regards,

Pandi Vijayakumar, Ph.D

Academic Editor

PLOS ONE

Additional Editor Comments (optional):

Reviewers' comments:

Reviewer's Responses to Questions

**Comments to the Author**

1. If the authors have adequately addressed your comments raised in a previous round of review and you feel that this manuscript is now acceptable for publication, you may indicate that here to bypass the “Comments to the Author” section, enter your conflict of interest statement in the “Confidential to Editor” section, and submit your "Accept" recommendation.

Reviewer #1: All comments have been addressed

Reviewer #2: All comments have been addressed

2. Is the manuscript technically sound, and do the data support the conclusions?

Reviewer #1: Yes

Reviewer #2: Yes

3. Has the statistical analysis been performed appropriately and rigorously? 

Reviewer #1: Yes

Reviewer #2: Yes

4. Have the authors made all data underlying the findings in their manuscript fully available?

Reviewer #1: Yes

Reviewer #2: Yes

5. Is the manuscript presented in an intelligible fashion and written in standard English?

Reviewer #1: Yes

Reviewer #2: No

6. Review Comments to the Author

Reviewer #1: (No Response)

Reviewer #2: (No Response)

7. PLOS authors have the option to publish the peer review history of their article (what does this mean?). If published, this will include your full peer review and any attached files.

Reviewer #1: No

Reviewer #2: No

---

## [Editor Report · Acceptance letter]

18 Jul 2022

PONE-D-22-11432R1 

Cryptanalysis and improved mutual authentication key agreement protocol using pseudo-identity 

Dear Dr. Pak:

I'm pleased to inform you that your manuscript has been deemed suitable for publication in PLOS ONE. Congratulations! Your manuscript is now with our production department. 

Kind regards, 

on behalf of

Dr. Pandi Vijayakumar 

Academic Editor

PLOS ONE